# Dynamic Graph Information Bottleneck

## ABSTRACT

Dynamic Graphs widely exist in the real world, which carry complicated spatial and temporal feature patterns, challenging their representation learning. Dynamic Graph Neural Networks (DGNNs) have shown impressive predictive abilities by exploiting the intrinsic dynamics. However, DGNNs exhibit limited robustness, prone to adversarial attacks. This paper presents the novel *Dynamic Graph Information Bottleneck (DGIB)* framework to learn robust and discriminative representations. Leveraged by the Information Bottleneck (IB) principle, we first propose the expected optimal representations should satisfy the *Minimal-Sufficient-Consensual (MSC)* Condition. To compress redundant as well as conserve meritorious information into latent representation, DGIB iteratively directs and refines the structural and feature information flow passing through graph snapshots. To meet the *MSC* Condition, we decompose the overall IB objectives into $\text{DGIB}_{MS}$ and $\text{DGIB}_C$, in which the $\text{DGIB}_{MS}$ channel aims to learn the minimal and sufficient representations, with the $\text{DGIB}_C$ channel guarantees the predictive consensus. Extensive experiments on real-world and synthetic dynamic graph datasets demonstrate the superior robustness of DGIB against adversarial attacks compared with state-of-the-art baselines in the link prediction task. To the best of our knowledge, DGIB is the first work to learn robust representations of dynamic graphs grounded in the information-theoretic IB principle.

## KEYWORDS

dynamic graph neural networks, robust representation learning, information bottleneck

## 1 INTRODUCTION

Dynamic graphs are prevalent in real-world scenarios, like the extensive structure of the World Wide Web [3, 4, 52] representing a vast dynamic network, and encompassing domains such as social networks [6, 16], financial transaction [38, 65], and traffic networks [30, 66], *etc.* Due to their intricate spatial and temporal correlation patterns, addressing a wide spectrum of applications like web link co-occurrence analysis [8, 22], relation prediction [24, 43], anomaly detection [7, 44] and traffic flow analysis [32, 57], *etc.* poses significant challenges. Leveraging their exceptional expressive capabilities, dynamic graph neural networks (DGNNs) [18, 50] intrinsically excel in the realm of dynamic graph representation learning by modeling both spatial and temporal predictive patterns, which is achieved with the combined merits of graph neural networks (GNNs)-based models and sequence-based models.

Recently, there has been an increasing emphasis on the efficacy enhancement of the DGNNs [13, 67, 68, 70], with a specific focus on augmenting their capabilities to capture intricate spatio-temporal feature patterns against the first-order Weisfeiler-Leman (1-WL) isomorphism test [36, 59]. However, most of the existing works still struggle with several challenges. One of the prominent challenges is brought by the over-smoothing phenomenon due to the message-passing mechanism in vanilla (D)GNNs [9]. Specifically, the inherent node features contain potentially irrelevant and spurious information, which will be aggregated over the edges, compromising the resilience of DGNNs, prone to ubiquitous noise of in-the-wild testing samples, and possible adversarial attacks.

To against adversarial attacks, the Information Bottleneck (IB) principle [54, 55] introduces an information-theoretic theory for robust representation learning, which encourages the model to acquire the most informative and predictive representation of the target, which satisfies both the minimal and sufficient assumption for optimal representations [12]. As illustrated in Figure 1(a), the IB principle serves to encourage the representation to capture maximum mutual information about the target and make accurate predictions (*Sufficient*). Simultaneously, it discourages the inclusion of redundant information from the input that is unrelated to predicting the target (*Minimal*). By adhering to this learning paradigm, the trained model naturally mitigates overfitting and becomes more robust to potential noise and adversarial attacks. However, directly applying IB to robust representation learning of dynamic graphs faces significant challenges. First, IB necessitates that the learning process adhere to the Markovian dependence, which requires data to be independent and identically distributed (*i.i.d.*). However, non-Euclidean dynamic graphs inherently do not satisfy the *i.i.d.* assumption. Second, dynamic graphs exhibit coupled structural and temporal features, where these discrete and intricate information flows make optimizing the IB objectives intractable.

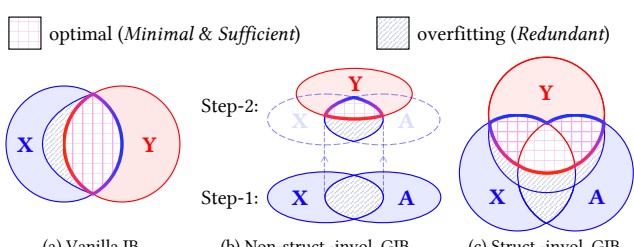

Figure 1: Comparison among different IB principles.

A few attempts have been made to extend the IB principle to *static* graphs [51, 60–64] by a two-step paradigm (Figure 1(b)), which initially obtains tractable representations by vanilla GNNs, and subsequently models the distributions of variables by variational inference [55], which enables the calculation of IB objectives. However, as the crucial structural feature information are absent from the direct IB optimization process, it leads to unsatisfying robust performance. To make structures straightforwardly involved in the IB optimization, [58] explicitly compresses the input from both the graph structure and node feature perspectives iteratively in a tractable and constrained searching space, which is then achieved by optimizing the estimated variational bounds of the IB objectives and leads to better robustness (Figure 1(c)). Accordingly, the learned representations satisfy the *MS* Assumption, which can reduce the

impact of structural and intricate feature noise, alleviate overfitting, and strengthen informative and discriminative prediction ability. To the best of our knowledge, there's no successful work to adapt the IB principle to *dynamic* graph representation learning with spatio-temporal graph structures directly involved.

Due to the local (single snapshot) and global (overall snapshots) joint reliance for prediction tasks on dynamic graphs, which introduce complex correlated spatio-temporal patterns evolving with time, extending the intractable IB principle to the dynamic graph representation learning is a non-trivial problem confronting the following major challenges:

- How to understand what constitutes the optimal representation that is both discriminative and robust for downstream task prediction under the dynamic scenario? (Section 4.1)
- How to appropriately compress the input dynamic graph features by optimizing the information flow across graph snapshots with structures straightforwardly involved? (Section 4.2)
- How to optimize the intractable IB objectives, which are incalculable on the non-Euclidean dynamic graphs? (Section 4.3)

**Present work.** To tackle the aforementioned challenges, we propose the innovative *Dynamic Graph Information Bottleneck (DGIB)* framework, aiming at striking a balance between the expressiveness and robustness of DGNNs. Our goal is to develop a robust dynamic graph representation learning framework with the theoretic support of the IB principle. To understand what contributes to the optimal representations that are both informative and robust for prediction in dynamic scenarios, we propose the concept of *Minimal-Sufficient-Consensual (MSC)* Condition with empirical and theoretical analysis on a case study. To conserve informative features in the input, we design the spatio-temporal sampling mechanism to derive the local dependence assumption, based on which we establish the DGIB principle that guides the information flow crossing graphs and iteratively refines structures and features (Figure 2(a)). Specifically, we decompose the overall DGIB objectives into $DGIB_{MS}$ and $DGIB_C$ channels, both sharing the same IB structures, cooperating to jointly satisfy the proposed *MSC* Condition (Figure 2(b)). To make the IB objectives tractable, we introduce appropriate variational upper and lower bounds for $DGIB_{MS}$ and $DGIB_C$, respectively, which guarantee rational estimation of the mutual information in DGIB objectives. We highlight the advantages of our DGIB as follows:

- We propose a novel framework named DGIB for robust dynamic graph representation learning with the information-theoretic support of the IB principle. To the best of our knowledge, this is the first exploration to extend IB on dynamic graphs with structures directly involved in IB optimization.
- We investigate the expected optimal representations for dynamic graphs and propose the *Minimal-Sufficient-Consensual (MSC)* Condition, which can be satisfied by the cooperation of both $DGIB_{MS}$ and $DGIB_C$ channels to refine the spatio-temporal information flow for feature compression. We further introduce their variational bounds to make training objectives tractable.
- Extensive experiments on both real-world and synthetic dynamic graph datasets demonstrate the superior robustness of our DGIB against targeted and non-targeted adversarial attacks compared with state-of-the-art baselines.

## 2 RELATED WORK

### 2.1 Dynamic Graph Representation Learning

Dynamic Graphs find applications in a wide variety of disciplines, including social networks, recommender systems, epidemiology, *etc.* Following [18, 50], dynamic graphs can be categorized into four levels based on their temporal granularity: static, edge-weighted, discrete, and continuous. Our primary concern lies in the representation learning for *discrete* dynamic graphs, which encompass multiple discrete graph snapshots arranged in chronological order.

Dynamic Graph Neural Networks (DGNNs) are widely adopted to learn dynamic graph representations by intrinsically modeling both spatial and temporal predictive patterns, which can be divided into two categories. **(1) *Stacked DGNNs*** employ separate GNNs to process each graph snapshot, and forward the output of each GNN to deep sequential models [19, 34], *etc.* Stacked DGNNs alternately model the dynamics to learn representations, which are the mainstream. **(2) *Integrated DGNNs*** function as encoders that combine GNNs and deep time-series models within a single layer, unifying spatial and temporal domain modeling. The deep sequential models are applied to initialize the weights of GNN layers.

However, despite efforts aimed at creating more powerful DGNNs with enhanced expressive capabilities, most of these models still lack robustness against adversarial attacks. Dynamic graphs, often derived from open data environments, inherently contain various forms of noise, and frequently redundant features unrelated to the prediction task, which can compromise DGNN performance in downstream tasks. Additionally, DGNNs are susceptible to inherent over-smoothing issues, making them sensitive to noise in real-world testing samples and vulnerable to adversarial attacks. Currently, no robust DGNN solutions have been effectively proposed.

### 2.2 Information Bottleneck

The Information Bottleneck (IB) principle aims to discover a concise code for the input signal while retaining the maximum information within the code for signal processing [54]. [55] initially extends Variational Information Bottleneck (VIB) to deep learning, named Deep VIB. Presently, IB and VIB are primarily associated with representation learning and feature compression. In representation learning, researchers employ either a deterministic or stochastic encoder to acquire a compact yet predictive representation of input data, facilitating diverse downstream applications in fields such as reinforcement learning [11, 21], computer vision [15, 31], natural language processing [40], *etc.* For feature compression, IB is employed to select a subset of input features, such as pixels in images or dimensions in embeddings that are maximally discriminative with respect to the target labels. Nevertheless, research on IB in the non-Euclidean dynamic graphs has been relatively limited, primarily due to the intractability of optimizing IB objectives.

There are some prior works that extend IB on *static* graphs, which can be categorized into two groups based on whether the graph structures are straightforwardly involved in the IB optimization process. **(1) *Non-structure-involved.*** These works follow the two-step learning paradigm, which firstly models latent representations, and consequently performs feature compression by IB, where the graph structural information is absent from the optimization process. For example, SIB [62–64] is proposed for the critical subgraph

recognition problem. HGIB [61] implements the consensus hypothesis of heterogeneous information networks in an unsupervised manner. VIB-GSL [51] first leverages IB to graph structure learning *etc.* To make IB objectives tractable, they model the distributions of representations by variational inference, which enables the IB terms computable for direct optimization. **(2) *Structure-involved.*** To make IB directly involved in the feature compression process, the pioneering GIB [58] explicitly extracts information with regularization of the structure and feature information in a tractable searching space, which follows a Markov Chain and is achieved by optimizing the estimated variational bounds of the IB objectives.

In conclusion, structure-involved GIBs demonstrate superior advantages over non-structure-involved ones by leveraging the significant graph structure patterns, which can reduce the impact of environment noise, enhance the robustness of models, as well as be informative and discriminative for downstream prediction tasks. However, extending the IB principle on dynamic graph learning proves challenging, as the impact of the spatio-temporal correlations on the Markov process should be elaborately considered.

## 3 NOTATIONS AND PRELIMINARIES

In this paper, random variables are denoted as **bold** letters while their realizations are *italic* letters. The ground-truth distribution is represented as $\mathbb{P}(\cdot)$, and $\mathbb{Q}(\cdot)$ denotes its approximation.

**Notation.** We primarily consider the *discrete* dynamic representation learning. A discrete dynamic graph can be denoted as a series of graphs snapshots $\mathbf{DG} = \{\mathcal{G}^t\}_{t=1}^T$, where $T$ is the time length. $\mathcal{G}^t = (\mathcal{V}^t, \mathcal{E}^t)$ is the graph at time $t$, where $\mathcal{V}^t$ is the node set and $\mathcal{E}^t$ is the edge set. Let $\mathbf{A}^t \in \{0,1\}^{N \times N}$ be the adjacency matrix and $\mathbf{X}^t \in \mathbb{R}^{N \times d}$ be the node features, where $N = |\mathcal{V}^t|$ denotes the number of nodes and $d$ denotes the feature dimensionality.

**Dynamic Graph Representation Learning.** As the most challenging task of dynamic graph representation learning, the future link prediction aims to train a model $f_{\theta} : \mathcal{V} \times \mathcal{V} \mapsto \{0,1\}^{N \times N}$ that predicts the existence of edges at $T+1$ given historical graphs $\mathcal{G}^{1:T}$ and next-step node features $\mathbf{X}^{T+1}$. Concretely, $f_{\theta} = w \circ g$ is compound of a DGNN $w(\cdot)$ to learn node representations and a link predictor $g(\cdot)$ for link prediction, *i.e.*, $\mathbf{Z}^{T+1} = w(\mathcal{G}^{1:T}, \mathbf{X}^{T+1})$ and $\hat{\mathbf{Y}}^{T+1} = g(\mathbf{Z}^{T+1})$. Our goal is to learn a robust representation against adversarial attacks with the optimal parameters $\theta^{\star}$.

**Information Bottleneck.** The Information Bottleneck (IB) principle trades off the data fit and robustness using mutual information (MI) as the cost function and regularizer. Given the input $\mathbf{X}$, representation $\mathbf{Z}$ of $\mathbf{X}$ and target $\mathbf{Y}$, the tuple $(\mathbf{X}, \mathbf{Y}, \mathbf{Z})$ follows the Markov Chain $< \mathbf{Y} \rightarrow \mathbf{X} \rightarrow \mathbf{Z} >$. IB learns the minimal and sufficient representation $\mathbf{Z}$ by optimizing the following objective:

$$\mathbf{Z} = \arg\min_{\mathbf{Z}} -I(\mathbf{Y}; \mathbf{Z}) + \beta I(\mathbf{X}; \mathbf{Z}), \qquad (1)$$

where $\beta$ is the Lagrangian parameter to balance the two terms. $I(\mathbf{X}; \mathbf{Y})$ represents the mutual information between the random variables $\mathbf{X}$ and $\mathbf{Y}$, which takes the form:

$$I(\mathbf{X}; \mathbf{Y}) = \mathrm{KL}[\mathbb{P}(\mathbf{X}, \mathbf{Y}) \parallel \mathbb{P}(\mathbf{X})\mathbb{P}(\mathbf{Y})], \qquad (2)$$

where $\mathrm{KL}[\cdot \parallel \cdot]$ is the Kullback-Liebler (KL) divergence [23], and $H(\cdot)$ denotes the information entropy.

## 4 DYNAMIC GRAPH INFORMATION BOTTLENECK

In this section, we elaborate on the proposed DGIB, where its principle and framework are shown in Figure 2. First, we propose the *Minimal-Sufficient-Consensual (MSC)* Condition that the expected optimal representations should satisfy. Then, we derive the DGIB principle by decomposing it into DGIB$_{MS}$ and DGIB$_C$ channels, which cooperate and contribute to satisfying the proposed *MSC* Condition. Lastly, we instantiate the DGIB principle with tractable variational bounds for efficient IB objective optimization.

## 4.1 DGIB Optimal Representation Condition

Given the input $\mathbf{X}$ and label $\mathbf{Y}$, the sufficient statistics theory [49] identifies the optimal representation of $\mathbf{X}$, namely, $S(\mathbf{X})$, which effectively encapsulates all the pertinent information contained within $\mathbf{X}$ concerning $\mathbf{Y}$, namely, $I(S(\mathbf{X}); \mathbf{Y}) = I(\mathbf{X}; \mathbf{Y})$. Steps further, the minimal and sufficient statistics $T(\mathbf{X})$ establish a fundamental sufficient partition of $\mathbf{X}$. This can be expressed through the Markov Chain: $MC_{\mathrm{IB}} < \mathbf{Y} \rightarrow \mathbf{X} \rightarrow S(\mathbf{X}) \rightarrow T(\mathbf{X}) >$, which holds true for a minimal sufficient statistics $T(\mathbf{X})$ in conjunction with any other sufficient statistics $S(\mathbf{X})$. Leveraging the Data Processing Inequality (DPI) [5], the *Minimal and Sufficient (MS)* Assumption satisfied representation can be optimized by:

$$T(\mathbf{X}) = \arg\min_{S(\mathbf{X}): I(S(\mathbf{X}); \mathbf{Y}) = I(\mathbf{X}; \mathbf{Y})} I(S(\mathbf{X}); \mathbf{X}), \qquad (3)$$

where the mapping $S(\cdot)$ can be relaxed to any encoder $\mathbb{P}(\mathbf{T} \mid \mathbf{X})$, which allows $S(\cdot)$ to capture as much as possible of $I(\mathbf{X}; \mathbf{Y})$.

**Case Study.** Non-structure-involved GIBs adapt the $MC_{\mathrm{IB}}$ to $MC_{\mathrm{GIB}} < \mathbf{Y} \rightarrow \mathcal{G} \rightarrow S(\mathcal{G}) \rightarrow \mathbf{Z} >$, where $\mathcal{G}$ contains both structures ($\mathbf{A}$) and node features ($\mathbf{X}$), leading to the minimal and sufficient $\mathbf{Z}$ with a balance between expressiveness and robustness. However, we observe a counterfactual result that if directly adapting $MC_{\mathrm{GIB}}$ to dynamic graphs as $MC_{\mathrm{DGIB}} :< \mathbf{Y}^{T+1} \rightarrow (\mathcal{G}^{1:T}, \mathbf{X}^{T+1}) \rightarrow S(\mathcal{G}^{1:T}, \mathbf{X}^{T+1}) \rightarrow \mathbf{Z}^{T+1} >$ and optimizing the overall objective:

$$\mathbf{Z}^{T+1} = \arg\min_{\mathbf{Z}^{T+1}} \left[ -I(\mathbf{Y}^{T+1}; \mathbf{Z}^{T+1}) + \beta I(\mathcal{G}^{1:T}, \mathbf{X}^{T+1}; \mathbf{Z}^{T+1}) \right] \quad (4)$$

will lead to sub-optimal *MS* representation $\hat{\mathbf{Z}}^{T+1}$.

Following settings in Appendix C.2, we adapt the GIB [58] to dynamic scenarios by processing each individual graph with original GIB [58] and then aggregating each output $\mathbf{Z}^t$ with LSTM [19] to learn the comprehensive representation $\hat{\mathbf{Z}}^{T+1}$, where the two steps are *jointly* optimized by Eq. (4). We evaluate the model performance of the link prediction task on the dynamic graph dataset ACT [29]. As depicted in Figure 3, the results reveal a noteworthy finding: the prediction performance achieved by $\hat{\mathbf{Z}}^{t+1}$ for the next time-step graph $\mathcal{G}^{t+1}$ during training significantly *surpasses* that of $\hat{\mathbf{Z}}^{T+1}$ for the target graph $\mathcal{G}^{T+1}$ during validating and testing.

We attribute the above results to the *laziness* of deep neural networks [33], leading to *non-consensual* feature compression for each time step. Specifically, there exist $T+1$ independent and identically distributed Markov Chains $MC_{\mathrm{DGIB}}^t < \mathbf{Y}^{t+1} \rightarrow (\mathcal{G}^t, \mathbf{X}^{t+1}) \rightarrow S(\mathcal{G}^t, \mathbf{X}^{t+1}) \rightarrow \mathbf{Z}^{t+1} >$. Optimizing each $MC_{\mathrm{DGIB}}^t$ as:

$$\mathbf{Z}^{t+1} = \arg\min_{\mathbf{Z}^{t+1}} \left[ -I(\mathbf{Y}^{t+1}; \mathbf{Z}^{t+1}) + \beta I(\mathcal{G}^t, \mathbf{X}^{t+1}; \mathbf{Z}^{t+1}) \right] \quad (5)$$

guarantees the learned $\mathbf{Z}^{t+1}$ meeting the *MS* Assumption respectively for each $\mathcal{G}^{t+1}$. However, as the spatio-temporal patterns

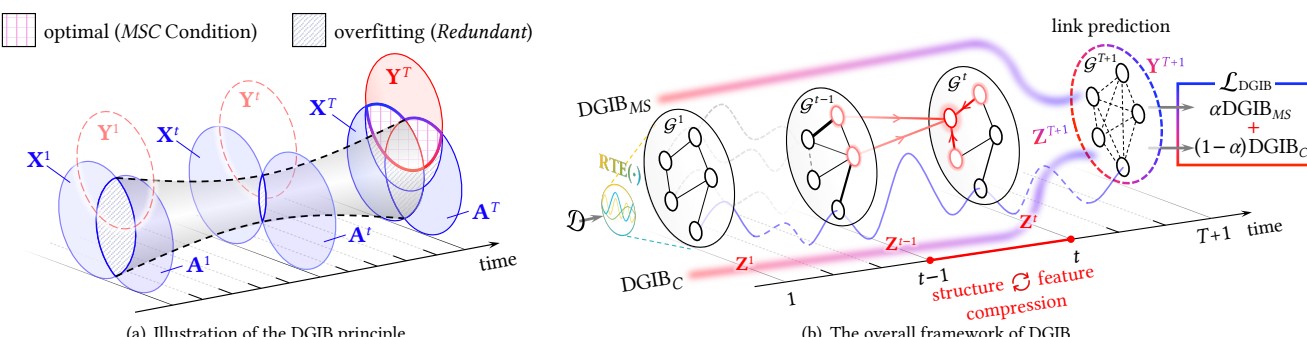

(a) Illustration of the DGIB principle.

(b) The overall framework of DGIB.

**Figure 2: The proposed DGIB principle and its overall framework. (a) simultaneously maximizing the mutual information between the representation and the target while constraining information with the input graphs. The significant graph structures are directly involved in the optimizing process. (b) iteratively compresses structures and node features between graphs. The overall $\mathcal{L}_{\text{DGIB}}$ is decomposed to DGIB$_{MS}$ and DGIB$_C$ channels, which act jointly to satisfy the *MSC* Condition.**

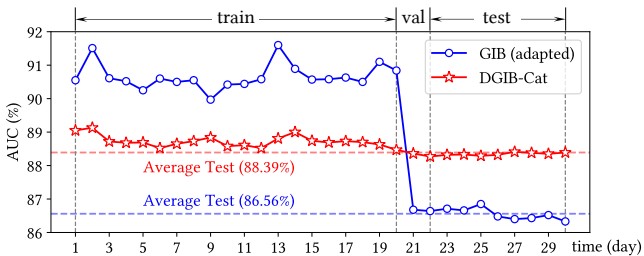

**Figure 3: Case study between adapted GIB and DGIB-Cat.**

among graph snapshots are intricately complicated, their dependencies are coupled. The adapted GIB tends to converge to a suboptimal *trivial* solution that satisfies each $MC^t_{\text{DGIB}}$, such that every $\hat{Z}^{t+1}$ is close to *MS* status for the next-step label $Y^{t+1}$, but fail to predict the final $Y^T$ due to its laziness. Consequently, the objective in Eq. (4) degrades to simply optimize the *union* of $\{\hat{Z}^t\}_{t=1}^{T+1}$.

The above analysis confirms the fact that the global representation $\hat{Z}^{T+1}$ does not align closely with the *MS* Assumption. To encourage $\hat{Z}^{T+1}$ to reach the *global* minimal and sufficient status, we apply an additional *Consensual* Constraint $C(\boldsymbol{\theta})$ on the sequence of $MC^t_{\text{DGIB}}$, which acts like the "baton" for compression process, and encourages conserving the consensual parts. We conclude the *Minimal-Sufficient-Consensual (MSC)* Condition as follows.

**Assumption 1** (Minimal-Sufficient-Consensual (MSC) Condition). Given $\mathbf{DG} = \{\mathcal{G}^t\}_{t=1}^T$, the optimal representation $Z^{T+1}$ for the robust future link prediction should satisfy the Minimal-Sufficient-Consensual (MSC) Condition, such that:

$$Z^{T+1} = \arg\min_{S(\mathcal{D}):I(S(\mathcal{D});Y^{T+1})=I(\mathcal{D};Y^{T+1})} I(S(\mathcal{D});\mathcal{D}), \quad (6)$$

where $\mathcal{D}$ is the training data containing previous graphs $\mathcal{G}^{1:T}$ and node feature $X^{T+1}$ at the next time-step, and $S(\cdot)$ is the partitions of $\mathcal{D}$, which is implemented by a stochastic encoder $\mathbb{P}(Z^{T+1} \mid \mathcal{D}, C(\boldsymbol{\theta}))$, where $C(\boldsymbol{\theta})$ satisfies the Consensual Constraint.

Assumption 1 declares the optimal representation $Z^{T+1}$ for the robust future link prediction task of dynamic graphs, which follows the Markovian dependence $MC_{\text{DGIB}}$, should be *Minimal*, *Sufficient* and *Consensual (MSC)*.

## 4.2 DGIB Principle Derivation

Prior to delving into the details of the DGIB principle, we first provide a formal definition of the Dynamic Information Bottleneck that satisfies *MSC* Condition.

**Definition 1** (Dynamic Graph Information Bottleneck). Given $\mathbf{DG} = \{\mathcal{G}^t\}_{t=1}^T$, and the nodes feature $X^{T+1}$ at the next time-step, the Dynamic Graph Information Bottleneck (DGIB) is to learn the optimal representation $Z^{T+1}$ that satisfies *MSC* Condition by:

$$Z^{T+1} = \arg\min_{\mathbb{P}(Z^{T+1}\mid\mathcal{D},C(\boldsymbol{\theta}))\in\Omega} \text{DGIB}(\mathcal{D}, Y^{T+1}; Z^{T+1})$$

$$\triangleq \left[-I(Y^{T+1}; Z^{T+1}) + \beta I(\mathcal{D}; Z^{T+1})\right], \quad (7)$$

where $\mathcal{D} = (\mathcal{G}^{1:T}, X^{T+1})$ is the input training data, and $\Omega$ defines the search space of the optimal DGNN model $\mathbb{P}(Z^{T+1} \mid \mathcal{D}, C(\boldsymbol{\theta}))$.

Compared with Eq. (4), Eq. (7) satisfies the additional *Consensual* Constraint $C(\boldsymbol{\theta})$ on $\mathbb{P}(Z^{T+1} \mid \mathcal{D})$ within the underlying search space $\Omega$. This encourages $Z^{T+1}$ to align more effectively with the *MSC* Condition. Subsequently, we decompose the overall DGIB into DGIB$_{MS}$ and DGIB$_C$ channels, both sharing the same IB structure as Eq. (1). DGIB$_{MS}$ aims to optimize $Z^{T+1}$ under $\mathbb{P}(Z^{T+1} \mid \mathcal{D}, \boldsymbol{\theta})$, while DGIB$_C$ encourages $Z^{1:T}$ and $Z^{T+1}$ share consensual predictive pattern for predcting $Y^{T+1}$ under $\mathbb{P}(Z^{T+1} \mid Z^{1:T}, C(\boldsymbol{\theta}))$. DGIB$_{MS}$ and DGIB$_C$ cooperate and contribute to satisfying the proposed *MSC* Condition under $\mathbb{P}(Z^{T+1} \mid Z^{1:T}, C(\boldsymbol{\theta}))$.

*4.2.1 Deriving the DGIB$_{MS}$ Principle.* In general, directly optimizing Eq. (7) poses a significant challenge due to the complex spatiotemporal correlations both among and within graphs. The *i.i.d.* assumption, typically necessary for estimating variational bounds, is a key factor in rendering the optimization of the IB objectives feasible. However, the *i.i.d.* assumption cannot be reached in dynamic graphs, as node features exhibit correlations owing to the underlying graph structures, and the dependencies are intricate. Consequently, approximating the optimal representation in the DGIB formulation appears unattainable without introducing additional assumptions. To solve the above challenges, we propose the specific and unique Spatio-Temporal Local Dependence Assumption following network science and probability theory [47].

**Assumption 2** (Spatio-Temporal Local Dependence). Given $\mathbf{DG} = \{\mathcal{G}^t\}_{t=1}^T$, let $\mathcal{N}_{ST}(v, k, t)$ be the spatio-temporal $k$-hop neighbors of any node $v \in \mathcal{V}$. The rest of the $\mathbf{DG}$ will be independent of node $v$ and its spatio-temporal $k$-hop neighbors, which takes the form:

$$\mathbb{P}\left(\mathbf{X}_v^t \mid \mathcal{G}_{\mathcal{N}_{ST}(v,k,t)}^{t-1:t}, \overline{\mathcal{G}_{\{v\}}^{1:T}}\right) = \mathbb{P}\left(\mathbf{X}_v^t \mid \mathcal{G}_{\mathcal{N}_{ST}(v,k,t)}^{t-1:t}\right), \quad (8)$$

where $\mathcal{G}_{\mathcal{N}_{ST}(v,k,t)}^{t-1:t}$ denotes $\mathcal{N}_{ST}(v, k, t)$ related subgraphs, and $\overline{\mathcal{G}_{\{v\}}^{1:T}}$ denotes complement graphs in terms of node $v$ and associated edges.

Assumption 2 is applied to constrain the search space $\Omega$ as $\mathbb{P}(\mathbf{Z}^{T+1} \mid \mathcal{D}, \boldsymbol{\theta})$, which leads to a more feasible $\text{DGIB}_{MS}$:

$$\mathbf{Z}^{T+1} = \arg \min_{\mathbb{P}(\mathbf{Z}^{T+1} \mid \mathcal{D}, \boldsymbol{\theta}) \in \Omega} \text{DGIB}_{MS}(\mathcal{D}, \mathbf{Y}^{T+1}; \mathbf{Z}^{T+1})$$

$$\triangleq \left[ - \underbrace{I(\mathbf{Y}^{T+1}; \mathbf{Z}^{T+1})}_{\text{Eq. (16)}} + \beta_1 \underbrace{I(\mathcal{D}; \mathbf{Z}^{T+1})}_{\text{Eq. (18)/Eq. (19), Eq. (20)}} \right]. \quad (9)$$

**Convolution Layer.** We assume the learning process follows the Markovian dependency in Figure 2. During each iteration $l$, the representations of each node $\mathbf{X}_v^t$ will be aggregated and updated by incorporating its spatio-temporal neighbors $\mathcal{N}_{ST}(v, k, t)$ on the refined structure $\hat{\mathbf{A}}^t$, which is adjusted to control the information flow across graph snapshots. In this way, $\text{DGIB}_{MS}$ requires to optimize the distributions of $\mathbb{P}(\hat{\mathbf{A}}^t \mid \hat{\mathbf{Z}}^t, \mathbf{Z}^{t-1}, \mathbf{A}^t)$ and $\mathbb{P}(\mathbf{Z}^t \mid \hat{\mathbf{Z}}^t, \mathbf{Z}^{t-1}, \hat{\mathbf{A}}^t)$.

**Variational Bounds of $\text{DGIB}_{MS}$.** We respectively introduce the lower bound of $I(\mathbf{Y}^{T+1}; \mathbf{Z}^{T+1})$ with respect to [42], and the upper bound of $I(\mathcal{D}; \mathbf{Z}^{T+1})$ inspired by [58], for Eq. (9).

**Proposition 1** (Lower Bound of $I(\mathbf{Y}^{T+1}; \mathbf{Z}^{T+1})$).

$$I(\mathbf{Y}^{T+1}; \mathbf{Z}^{T+1}) \geq 1 + \mathbb{E}_{\mathbb{P}(\mathbf{Y}^{T+1}, \mathbf{Z}^{T+1})} \left[ \log \frac{\mathbb{Q}_1(\mathbf{Y}^{T+1} \mid \mathbf{Z}^{T+1})}{\mathbb{Q}_2(\mathbf{Y}^{T+1})} \right]$$

$$- \mathbb{E}_{\mathbb{P}(\mathbf{Y}^{T+1})\mathbb{P}(\mathbf{Z}^{T+1})} \left[ \frac{\mathbb{Q}_1(\mathbf{Y}^{T+1} \mid \mathbf{Z}^{T+1})}{\mathbb{Q}_2(\mathbf{Y}^{T+1})} \right]. \quad (10)$$

**Proposition 2** (Upper Bound of $I(\mathcal{D}; \mathbf{Z}^{T+1})$). Let $\mathcal{I}_A, \mathcal{I}_Z \subset [T+1]$ be the random time indices sets. Based on the Markov property $\mathcal{D} \perp \mathbf{Z}^{T+1} \mid (\{\hat{\mathbf{A}}^t\}_{t \in \mathcal{I}_A} \cup \{\mathbf{Z}^t\}_{t \in \mathcal{I}_Z})$, for any $\mathbb{Q}(\hat{\mathbf{A}}^t)$ and $\mathbb{Q}(\mathbf{Z}^t)$:

$$I(\mathcal{D}; \mathbf{Z}^{T+1}) \leq I(\mathcal{D}; \{\hat{\mathbf{A}}^t\}_{t \in \mathcal{I}_A} \cup \{\mathbf{Z}^t\}_{t \in \mathcal{I}_Z}) \leq \sum_{t \in \mathcal{I}_A} \mathcal{A}^t + \sum_{t \in \mathcal{I}_Z} \mathcal{Z}^t, \quad (11)$$

$$where \quad \mathcal{A}^t = \mathbb{E} \left[ \log \frac{\mathbb{P}(\hat{\mathbf{A}}^t \mid \hat{\mathbf{Z}}^t, \mathbf{Z}^{t-1}, \mathbf{A}^t)}{\mathbb{Q}(\hat{\mathbf{A}}^t)} \right], \quad (12)$$

$$\mathcal{Z}^t = \mathbb{E} \left[ \log \frac{\mathbb{P}(\mathbf{Z}^t \mid \hat{\mathbf{Z}}^t, \mathbf{Z}^{t-1}, \hat{\mathbf{A}}^t)}{\mathbb{Q}(\mathbf{Z}^t)} \right]. \quad (13)$$

Proofs for Proposition 1 and 2 are provided in Appendix B.

*4.2.2 Deriving the $\text{DGIB}_C$ Principle.* To ensure $\mathbf{Z}^{T+1}$ adheres to the *Consensual* Constraint with respect to $\mathbf{Z}^{1:T}$, we further constrain the search space $\Omega$ as $\mathbb{P}(\mathbf{Z}^{T+1} \mid \mathbf{Z}^{1:T}, C(\boldsymbol{\theta}))$ and optimizing $\text{DGIB}_C$:

$$\mathbf{Z}^{T+1} = \arg \min_{\mathbb{P}(\mathbf{Z}^{T+1} \mid \mathbf{Z}^{1:T}, C(\boldsymbol{\theta})) \in \Omega} \text{DGIB}_C(\mathbf{Z}^{1:T}, \mathbf{Y}^{T+1}; \mathbf{Z}^{T+1})$$

$$\triangleq \left[ - \underbrace{I(\mathbf{Y}^{T+1}; \mathbf{Z}^{T+1})}_{\text{Eq. (16)}} + \beta_2 \underbrace{I(\mathbf{Z}^{1:T}; \mathbf{Z}^{T+1})}_{\text{Eq. (21)}} \right]. \quad (14)$$

**Variational Bounds of $\text{DGIB}_C$.** The lower bound of $I(\mathcal{D}; \mathbf{Z}^{T+1})$ in Eq. (14) is consistent with Proposition 1. We introduce the upper bound of $I(\mathbf{Z}^{1:T}; \mathbf{Z}^{T+1})$, which is proved in Appendix B.

**Proposition 3** (Upper Bound of $I(\mathbf{Z}^{1:T}; \mathbf{Z}^{T+1})$).

$$I(\mathbf{Z}^{1:T}; \mathbf{Z}^{T+1}) \leq \mathbb{E} \left[ \log \frac{\mathbb{P}(\mathbf{Z}^{T+1} \mid \mathbf{Z}^{1:T})}{\mathbb{Q}(\mathbf{Z}^{T+1})} \right]. \quad (15)$$

The distinctions between $\text{DGIB}_{MS}$ and $\text{DGIB}_C$ primarily pertain to their *inputs* and *search space*. $\text{DGIB}_{MS}$ utilizes the original training data $\mathcal{D} = (\mathcal{G}^{1:T}, \mathbf{X}^{T+1})$ with $\mathbb{P}(\mathbf{Z}^{T+1} \mid \mathcal{D}, \boldsymbol{\theta})$ constraint as input, whereas $\text{DGIB}_C$ takes intermediate variables $\mathbf{Z}^{1:T}$ as input with $\mathbb{P}(\mathbf{Z}^{T+1} \mid \mathbf{Z}^{1:T}, C(\boldsymbol{\theta}))$ constraint. $\text{DGIB}_{MS}$ and $\text{DGIB}_C$ mutually constrains each other via the optimization process, ultimately leading to the satisfaction of the *MSC* Condition approved by $\mathbf{Z}^{T+1}$.

### 4.3 DGIB Principle Instantiation

To jointly optimize $\text{DGIB}_{MS}$ and $\text{DGIB}_C$, we begin by specifying the lower and upper bounds defined in Proposition 1, 2 and 3.

**Instantiation for Eq. (10).** To specify the lower bound, we set $\mathbb{Q}_1(\mathbf{Y}^{T+1} \mid \mathbf{Z}^{T+1}) = \text{Cat}(g(\mathbf{Z}^{T+1}))$, where $\text{Cat}(\cdot)$ denotes the categorical distribution, and set $\mathbb{Q}_2(\mathbf{Y}^{T+1}) = \mathbb{P}(\mathbf{Y}^{T+1})$. As the second term in Eq. (10) empirically converges to 1, we ignore it. Thus, the RHS of Eq. (10) reduces to the cross-entropy loss [42], *i.e.*:

$$I(\mathbf{Y}^{T+1}; \mathbf{Z}^{T+1}) \doteq -\mathcal{L}_{\text{CE}}(g(\mathbf{Z}^{T+1}), \mathbf{Y}^{T+1}). \quad (16)$$

**Instantiation for Eq. (12).** We instantiate DGIB on the backbone of the GAT [56], where the attentions can be utilized to refine the initial graph structure or as the parameters of the neighbor sampling distributions. Concretely, let $\phi_{v,k}^t$ as the logits of the attention weights between node $v$ and its spatio-temporal neighbors. We assume the prior of $\mathbb{P}(\hat{\mathbf{A}}^t \mid \hat{\mathbf{Z}}^t, \mathbf{Z}^{t-1}, \mathbf{A}^t)$ follows the Bernoulli distribution $\text{Bern}(\cdot)$ or the categorical distribution $\text{Cat}(\cdot)$ both parameterized by $\phi_{v,k}^t$. We assume $\mathbb{Q}(\hat{\mathbf{A}}^t)$ respectively follows the non-informative $\text{Bern}(\cdot)$ or $\text{Cat}(\cdot)$ parameterized by certain constants. Thus, the RHS of Eq. (12) is estimated by:

$$\mathcal{A}^t \doteq \mathbb{E}_{\mathbb{P}(\hat{\mathbf{A}}^t \mid \hat{\mathbf{Z}}^t, \mathbf{Z}^{t-1}, \mathbf{A}^t)} \left[ \log \frac{\mathbb{P}(\hat{\mathbf{A}}^t \mid \hat{\mathbf{Z}}^t, \mathbf{Z}^{t-1}, \mathbf{A}^t)}{\mathbb{Q}(\hat{\mathbf{A}}^t)} \right], \quad (17)$$

which can be further instantiated as:

$$\mathcal{A}_{\text{B}}^t \doteq \sum_{v \in \mathcal{V}^t} \text{KL} \left[ \text{Bern}(\phi_{v,k}^t) \middle\| \text{Bern}(|\mathcal{N}_{ST}(v, k, t)|^{-1}) \right], \quad (18)$$

$$\text{or} \quad \mathcal{A}_{\text{C}}^t \doteq \sum_{v \in \mathcal{V}^t} \text{KL} \left[ \text{Cat}(\phi_{v,k}^t) \middle\| \text{Cat}(|\mathcal{N}_{ST}(v, k, t)|^{-1}) \right]. \quad (19)$$

**Instantiation for Eq. (13) and Eq. (15).** To estimate $\mathcal{Z}^t$, we set the prior distribution of $\mathbb{P}(\mathbf{Z}^t \mid \hat{\mathbf{Z}}^t, \mathbf{Z}^{t-1}, \hat{\mathbf{A}}^t)$ follows a multivariate normal distribution $N(\mathbf{Z}^t; \boldsymbol{\mu}_{\mathbb{P}}, \sigma_{\mathbb{P}}^2)$, while $\mathbb{Q}(\mathbf{Z}^t) \sim N(\mathbf{Z}^t; \boldsymbol{\mu}_{\mathbb{Q}}, \sigma_{\mathbb{Q}}^2)$. Inspired by the Markov Chain Monte Carlo (MCMC) sampling [14], Eq. (13) can be estimated with sampled $\mathcal{V}^{t\prime}$:

$$\mathcal{Z}^t \doteq \sum_{v \in \mathcal{V}^{t\prime}} \left[ \log \Phi(\mathbf{Z}_v^t; \boldsymbol{\mu}_{\mathbb{P}}, \sigma_{\mathbb{P}}^2) - \log \Phi(\mathbf{Z}_v^t; \boldsymbol{\mu}_{\mathbb{Q}}, \sigma_{\mathbb{Q}}^2) \right], \quad (20)$$

where $\Phi(\cdot)$ is the Probability Density Function (PDF) of the normal distribution. Similarly, we specify Eq. (15) with sampled $\mathcal{V}^{(T+1)\prime}$:

$$\sum_{v \in \mathcal{V}^{(T+1)\prime}} \left[ \log \Phi(\mathbf{Z}_v^{T+1}; \boldsymbol{\mu}_{\mathbb{P}}, \sigma_{\mathbb{P}}^2) - \log \Phi(\mathbf{Z}_v^{T+1}; \boldsymbol{\mu}_{\mathbb{Q}}, \sigma_{\mathbb{Q}}^2) \right]. \quad (21)$$

**Training Objectives.** To acquire a tractable version of Eq. (7), we first plug Eq. (16), Eq. (18)/Eq. (19) and Eq. (20) into Eq. (9), respectively, to estimate $\text{DGIB}_{MS}$. Then plug Eq. (16) and Eq. (21) into Eq. (14) to estimate $\text{DGIB}_C$. The overall training objectives of the proposed DGIBcan be rewritten as:

$$\mathcal{L}_{\text{DGIB}} = \alpha\text{DGIB}_{MS} + (1 - \alpha)\text{DGIB}_C, \tag{22}$$

where the $\alpha$ is a trade-off hyperparameter.

## 4.4 Optimization and Complexity Analysis

The overall training pipeline of DGIB is shown in Algorithm 1. With the proposed upper and lower bounds for intractable terms in both $\text{DGIB}_{MS}$ and $\text{DGIB}_C$, the overall framework can be trained end-to-end using back-propagation, and thus we can use gradient descent to optimize. Based on the detailed analysis in Appendix A, the time complexity of our method is:

$$O(2Lk(T + 1)|\mathcal{V}||\mathcal{E}|dd'^2) + O(|\mathcal{E}'|d'), \tag{23}$$

which is on par with the state-of-the-art DGNNs. We further illustrate the training time efficiency in Appendix C.5.

## 5 EXPERIMENT

In this section, we conduct extensive experiments on both real-world and synthetic dynamic graph datasets to evaluate the robustness of our DGIB against adversarial attacks. We first introduce the experimental settings and then present the results. Additional configurations and results can be found in Appendix C and D.

## 5.1 Experimental Settings

*5.1.1 Dynamic Graph Datasets.* In order to comprehensively evaluate the effectiveness of our proposed method, we use three real-world dynamic graph datasets to evaluate DGIB[1] on the challenging future link prediction task. **COLLAB** [53] is an academic collaboration dataset with papers published in 16 years, which reveals the dynamic citation networks among authors. **Yelp** [46] contains customer reviews on business for 24 months, which are collected from the crowd-sourced local business review and social networking web. **ACT** [29] describes the actions taken by users on a popular MOOC website within 30 days, and each action has a binary label. Statistics of the datasets are concluded in Table B.1, which also contains the split of snapshots for training, validation, and testing.

*5.1.2 Baselines.* We compare DGIB with three categories baselines.

- Static GNNs: **GAE** [27] and **VGAE** [27] are representative GCN [26] based autoencoders on static graphs, which are widely used for the link prediction task. **GAT** [56] using attention mechanisms to dynamically weight and aggregate node features, which is also the default backbone of DGIB.
- Dynamic GNNs: **GCRN** [48] adopts GCNs to obtain node embeddings, followed by a GRU [10] to capture temporal relations. **EvolveGCN** [41] applies an LSTM [20] or GRU to evolve the parameters of GCNs. **DySAT** [46] models self-attentions in both structural and temporal domains.
- Robust and Generalized (D)GNNs: **IRM** [2] learns robust representation by minimizing invariant risk. **V-REx** [28] extends

---

[1]The code of DGIB is available at  https://anonymous.4open.science/r/DGIB.

IRM [2] by reweighting the risk. **GroupDRO** [45] reduces the risk gap across training distributions. **RGCN** [69] fortifies GCNs against adversarial attacks by Gaussian reparameterization and variance-based attention. **DIDA** [67] exploits robust and generalized predictive patterns on dynamic graphs. **GIB** [58] is the most relevant baseline to ours, which learns robust representations with structure-involved IB principle.

*5.1.3 Adversarial Attack Settings.* We compare baselines and the proposed DGIB under two adversarial attack settings.

- Non-targeted Adversarial Attack: We produce synthetic datasets by attacking *graph structures* and *node features*, respectively. **(1) *Attack graph structures.*** We randomly remove one out of five types of links in training and validation graphs in each dataset (information on link type has been removed after the above operations), which is more practical and challenging in real-world scenarios as the model cannot get access to any features about the filtered links. **(2) *Attack node features.*** We add random Gaussian noise $\lambda \cdot r \cdot \epsilon$ to each dimension of the node features for all nodes, where $r$ is the reference amplitude of original features, and $\epsilon \sim N(0, 1)$. $\lambda \in \{0.5, 1.0, 1.5\}$ acts as the parameter to control the attacking degree.
- Targeted Adversarial Attack: We apply the prevailing NET-TACK [71], a strong targeted adversarial attack library on graphs designed to target nodes by altering their connected edges or node features. We simultaneously consider the *evasive* attack and *poisoning* attack. **(1) *Evasive attack.*** We train the model on clean datasets and perform attacking on each graph snapshot in the testing split. **(2) *Poisoning attack.*** We attack the whole dataset before model training and testing. In both scenarios, we follow the default settings of NETTACK [71] to select targeted attacking nodes, and choose GAT [56] as the surrogate model with default parameter settings. We set the number of perturbations $n$ in $\{1, 2, 3, 4\}$.

*5.1.4 Hyperparameter Settings.* We set the number of layers as two for baselines as suggested, and as one for DGIB to avoid overfitting. We set the representation dimension of all baselines and our DGIB to be 128. The hyperparameters of baselines are set as the suggested value in their papers or carefully tuned for fairness. The suggested values of $\alpha$, $\beta_1$ and $\beta_2$ can be found in the configuration files. For the optimization, we use Adam [25] with a learning rate selected from {1e-03, 1e-04, 1e-05, 1e-06} adopt the grid search for the best performance using the validation split. We set the maximum epoch number as 1000 with the early stopping mechanism.

## 5.2 Against Non-targeted Adversarial Attacks

In this section, we evaluate model performance on the future link prediction task, as well as the robustness against non-targeted adversarial attacks in terms of graph structures and node features. Specifically, we train baselines and our DGIB on the clean datasets, after which we perturb edges and features respectively on the testing split following the experimental settings. Note that, DGIB with a prior of the Bernoulli distribution as Eq. (18) is referred to as DGIB-Bern, while DGIB with a prior of the categorical distribution in Eq. (19) is denoted as DGIB-Cat. Results are reported with the metric of AUC (%) score in five runs and concluded in Table 1.

**Table 1: AUC score (% ± standard deviation) of future link prediction task on real-world datasets against *non-targeted* adversarial attacks. The best results are shown in bold type and the runner-ups are underlined.**

| Dataset | COLLAB | | | | | Yelp | | | | | ACT | | | | |
|---|---|---|---|---|---|---|---|---|---|---|---|---|---|---|---|
| Model | Clean | Structure Attck | Feature Attack | | | Clean | Structure Attck | Feature Attack | | | Clean | Structure Attck | Feature Attack | | |
| | | | $\lambda = 0.5$ | $\lambda = 1.0$ | $\lambda = 1.5$ | | | $\lambda = 0.5$ | $\lambda = 1.0$ | $\lambda = 1.5$ | | | $\lambda = 0.5$ | $\lambda = 1.0$ | $\lambda = 1.5$ |
| GAE [27] | 77.15±0.5 | 74.04±0.8 | 50.59±0.8 | 44.66±0.8 | 43.12±0.8 | 70.67±1.1 | 64.45±5.0 | 51.05±0.6 | 45.41±0.6 | 41.56±0.9 | 72.31±0.5 | 60.27±0.4 | 56.56±0.5 | 52.52±0.6 | 50.36±0.9 |
| VGAE [27] | 86.47±0.0 | 74.95±1.2 | 56.75±0.6 | 50.39±0.7 | 48.68±0.7 | 76.54±0.5 | 65.33±1.4 | 55.53±0.7 | 49.88±0.8 | 45.08±0.6 | 79.18±0.5 | 66.29±1.3 | 60.67±0.7 | 57.39±0.8 | 55.27±1.0 |
| GAT [56] | 88.26±0.4 | 77.29±1.8 | 58.13±0.9 | 51.41±0.9 | 49.77±0.9 | 77.93±0.1 | 69.35±1.6 | 56.72±0.3 | 52.51±0.5 | 46.21±0.5 | 85.07±0.3 | 77.55±1.2 | 66.05±0.4 | 61.85±0.3 | 59.05±0.3 |
| GCRN [48] | 82.78±0.5 | 69.72±0.5 | 54.07±0.9 | 47.78±0.8 | 46.18±0.9 | 68.59±1.0 | 54.68±7.6 | 52.68±0.6 | 46.85±0.6 | 40.45±0.6 | 76.28±0.5 | 64.35±1.2 | 59.48±0.7 | 54.16±0.6 | 53.88±0.7 |
| EvolveGCN [41] | 86.62±1.0 | 76.15±0.9 | 56.82±1.2 | 50.33±1.0 | 48.55±1.0 | 78.21±0.0 | 53.82±2.0 | 57.91±0.5 | 51.82±0.3 | 45.32±1.0 | 74.55±0.3 | 63.17±1.0 | 61.02±0.5 | 53.34±0.5 | 51.62±0.7 |
| DySAT [46] | 88.77±0.2 | 76.59±0.2 | 58.28±0.3 | 51.52±0.3 | 49.32±0.5 | 78.87±0.6 | 66.09±1.4 | 58.46±0.4 | 52.33±0.7 | 46.24±0.7 | 78.52±0.4 | 66.55±1.2 | 61.94±0.8 | 56.98±0.4 | 54.14±0.7 |
| IRM [2] | 87.96±0.9 | 75.42±0.9 | 60.51±1.3 | 53.89±1.1 | 52.17±0.9 | 66.49±10.8 | 56.02±16.0 | 50.96±3.3 | 48.58±5.2 | 45.32±3.3 | 80.02±0.6 | 69.19±1.4 | 62.84±0.1 | 57.28±0.2 | 56.04±0.2 |
| V-REx [27] | 88.31±0.3 | 76.24±0.3 | 61.23±1.5 | 54.51±1.0 | 52.24±1.1 | 79.04±0.2 | 66.41±1.9 | 61.49±0.5 | 53.72±1.0 | 51.32±0.9 | 83.11±0.3 | 70.15±1.1 | 65.59±0.1 | 60.03±0.3 | 58.79±0.2 |
| GroupDRO [45] | 88.76±0.1 | 76.33±0.3 | 61.10±1.3 | 54.62±1.0 | 52.33±0.8 | 79.38±0.4 | 66.97±0.6 | 61.78±0.8 | 55.37±0.9 | 52.18±0.7 | 85.19±0.5 | 74.35±1.6 | 66.05±0.5 | 61.85±0.4 | 59.05±0.3 |
| RGCN [69] | 88.21±0.1 | 78.66±0.7 | 61.29±0.5 | 54.29±0.6 | 52.99±0.6 | 77.28±0.3 | 74.29±0.4 | 59.72±0.3 | 52.88±0.3 | 50.40±0.2 | 87.22±0.2 | 82.66±0.4 | 68.51±0.2 | 62.67±0.2 | 61.31±0.2 |
| DIDA [67] | 91.97±0.0 | 80.87±0.4 | 61.32±0.8 | 55.77±0.9 | 54.91±0.9 | 78.22±0.4 | 75.92±0.9 | 60.83±0.6 | 54.11±0.6 | 50.21±0.6 | 89.84±0.8 | 78.64±1.0 | 70.97±0.2 | 64.49±0.4 | 62.57±0.2 |
| GIB [58] | 91.36±0.2 | 80.89±0.1 | 61.88±0.8 | 55.15±0.8 | 54.65±0.9 | 77.52±0.4 | 75.03±0.3 | 61.94±0.9 | 56.15±0.3 | 52.21±0.8 | 92.33±0.3 | 86.99±0.3 | 72.16±0.5 | 66.72±0.2 | 64.96±0.5 |
| **DGIB-Bern** | 92.17±0.2 | 83.58±0.1 | 63.54±0.9 | 56.92±1.0 | **56.24±1.0** | 76.88±0.2 | 75.61±0.0 | **63.91±0.9** | **59.28±0.9** | **54.77±0.95** | 94.49±0.2 | 87.75±0.1 | 73.05±0.9 | 68.49±0.9 | **66.27±0.9** |
| **DGIB-Cat** | **92.68±0.1** | **84.16±0.1** | **63.99±0.5** | **57.76±0.8** | 55.63±1.0 | **79.53±0.2** | **77.72±0.1** | 61.42±0.9 | 55.12±0.7 | 51.90±0.9 | **94.89±0.2** | **88.27±0.2** | **73.92±0.8** | **68.88±0.9** | 65.99±0.7 |

**Table 2: AUC score (% ± standard deviation) of future link prediction task on real-world datasets against *targeted* adversarial attacks. The best results are shown in bold type and the runner-ups are underlined.**

| Dataset | Model | Clean | Evasive Attack | | | | | Poisoning Attack | | | | |
|---|---|---|---|---|---|---|---|---|---|---|---|---|
| | | | $n = 1$ | $n = 2$ | $n = 3$ | $n = 4$ | Avg. Decrease | $n = 1$ | $n = 2$ | $n = 3$ | $n = 4$ | Avg. Decrease |
| COLLAB | VGAE [27] | 86.47±0.0 | 73.39±0.1 | 62.18±0.1 | 51.72±0.1 | 46.97±0.1 | ↓ 32.27 | 63.42±0.3 | 52.63±0.3 | 50.98±0.4 | 45.64±0.3 | ↓ 38.51 |
| | GAT [56] | 88.26±0.4 | 76.21±0.1 | 66.56±0.1 | 57.92±0.1 | 50.96±0.1 | ↓ 28.71 | 66.59±0.5 | 55.31±0.6 | 51.34±0.7 | 48.99±0.9 | ↓ 37.05 |
| | DySAT [46] | 88.77±0.2 | 77.91±0.1 | 68.22±0.1 | 58.82±0.1 | 51.39±0.1 | ↓ 27.80 | 69.02±0.3 | 57.62±0.3 | 52.76±0.3 | 50.07±0.8 | ↓ 35.37 |
| | RGCN [69] | 88.21±0.1 | 77.65±0.1 | 67.11±0.1 | 59.06±0.1 | 52.02±0.1 | ↓ 27.49 | 69.48±0.2 | 58.39±0.3 | 52.48±0.6 | 50.62±0.9 | ↓ 34.53 |
| | GIB [58] | 91.36±0.2 | 78.95±0.0 | 69.63±0.1 | 60.98±0.0 | 54.48±0.2 | ↓ 27.74 | 71.47±0.3 | 61.03±0.4 | 54.97±0.7 | 52.09±1.0 | ↓ 34.44 |
| | **DGIB-Bern** | 92.17±0.2 | **81.36±0.0** | **72.79±0.0** | **63.25±0.1** | **57.22±0.1** | ↓ **25.51** | **74.06±0.3** | **61.93±0.2** | **56.57±0.2** | 52.62±0.3 | ↓ **33.49** |
| | **DGIB-Cat** | **92.68±0.1** | 81.29±0.0 | 71.32±0.1 | 62.03±0.1 | 55.08±0.1 | ↓ 27.24 | 72.55±0.2 | 60.99±0.3 | 55.62±0.4 | **53.08±0.3** | ↓ 34.65 |
| Yelp | VGAE [27] | 76.54±0.5 | 65.86±0.1 | 54.82±0.2 | 48.08±0.1 | 46.25±0.1 | ↓ 29.77 | 62.73±0.6 | 52.61±0.4 | 47.72±0.4 | 45.43±0.5 | ↓ 31.90 |
| | GAT [56] | 77.93±0.1 | 67.96±0.1 | 59.47±0.1 | 50.27±0.1 | 48.62±0.1 | ↓ 27.39 | 65.34±0.5 | 54.51±0.2 | 50.24±0.4 | 48.96±0.4 | ↓ 29.72 |
| | DySAT [46] | 78.87±0.6 | 69.77±0.1 | 60.66±0.1 | 52.16±0.1 | 50.15±0.1 | ↓ 26.22 | 66.87±0.6 | 56.31±0.3 | 50.44±0.6 | 50.49±0.5 | ↓ 28.96 |
| | RGCN [69] | 77.28±0.3 | 68.54±0.1 | 60.69±0.1 | 51.51±0.1 | 49.72±0.1 | ↓ 25.44 | 65.55±0.4 | 55.47±0.3 | 49.08±0.6 | 49.09±0.6 | ↓ 29.09 |
| | GIB [58] | 77.52±0.4 | 68.59±0.1 | 61.22±0.1 | 51.26±0.1 | 49.58±0.1 | ↓ 25.61 | 65.59±0.3 | 56.79±0.3 | 50.92±0.4 | 49.55±0.4 | ↓ 28.13 |
| | **DGIB-Bern** | 76.88±0.2 | **72.27±0.1** | 60.96±0.0 | **54.32±0.1** | **51.73±0.1** | ↓ **22.19** | **68.64±0.2** | 56.73±0.2 | **53.18±0.3** | 50.21±0.2 | ↓ **25.61** |
| | **DGIB-Cat** | **79.53±0.2** | 70.17±0.0 | **62.25±0.1** | 52.69±0.1 | 50.87±0.1 | ↓ 25.82 | 67.38±0.3 | **57.02±0.2** | 51.39±0.2 | **50.53±0.2** | ↓ 28.85 |
| ACT | VGAE [27] | 79.18±0.5 | 67.59±0.1 | 62.98±0.1 | 54.33±0.1 | 52.26±0.0 | ↓ 25.11 | 62.55±1.6 | 55.15±1.7 | 51.02±1.8 | 50.11±1.9 | ↓ 30.90 |
| | GAT [56] | 85.07±0.3 | 75.14±0.1 | 67.25±0.1 | 59.75±0.1 | 58.51±0.1 | ↓ 23.40 | 71.26±0.9 | 61.43±1.1 | 57.35±1.1 | 58.53±1.0 | ↓ 26.95 |
| | DySAT [46] | 78.52±0.4 | 70.64±0.1 | 63.35±0.0 | 56.36±0.0 | 55.12±0.1 | ↓ 21.84 | 66.21±0.9 | 56.28±0.9 | 53.45±1.1 | 54.43±1.0 | ↓ 26.65 |
| | RGCN [69] | 87.22±0.2 | 78.64±0.1 | 70.11±0.1 | 62.99±0.1 | 61.31±0.1 | ↓ 21.73 | 73.71±0.8 | 63.43±0.9 | 59.97±1.3 | 60.41±0.8 | ↓ **26.18** |
| | GIB [58] | 92.33±0.3 | 85.61±0.1 | 74.08±0.1 | 65.44±0.1 | 64.04±0.1 | ↓ 21.70 | 80.01±0.7 | 67.04±0.8 | 63.85±0.6 | 60.95±0.7 | ↓ 26.39 |
| | **DGIB-Bern** | 94.49±0.2 | **89.83±0.1** | **85.81±0.1** | **79.95±0.1** | **78.01±0.1** | ↓ **11.73** | **80.92±0.3** | **70.76±0.4** | **65.27±0.6** | 61.93±0.9 | ↓ 26.21 |
| | **DGIB-Cat** | **94.89±0.2** | 84.98±0.1 | 76.78±0.1 | 67.69±0.1 | 66.68±0.1 | ↓ 21.98 | 80.16±0.4 | 68.71±0.5 | 64.38±0.6 | **65.43±0.9** | ↓ 26.57 |

**Results**. DGIB-Bern and DGIB-Cat outperform GIB [58] and other baselines in most scenarios. Particularly, under structure attacks, DGIB-Bern improves by 9.4%, 15.1%, and 22.4% for each dataset, while 10.1%, 18.3%, and 23.1% for DGIB-Cat. Under feature attacks, DGIB-Bern improves by an average of 9.9%, 14.0%, and 15.1% for each dataset, while 10.3%, 7.9%, and 15.6% for DGIB-Cat. Static GNN baselines fail in all settings as they cannot model dynamics. Dynamic GNN baselines fail due to their weak robustness against adversarial attacks. Some robust and generalized (D)GNNs, such as GroupDRO [45], DIDA [67] and GIB [58] outperform DGIB-Bern or DGIB-Cat slightly in a few scenarios, but they generally fall behind DGIB in most cases due to the insufficient and inaccurate feature compression and conservation, which greatly impact the model robustness against the non-targeted adversarial attacks.

## 5.3 Against Targeted Adversarial Attacks

In this section, we continue to compare the proposed DGIB with competitive baselines standing out in Table 1, considering the link prediction performance and robustness against targeted adversarial attacks, which reveals whether we successfully defended the attacks. Specifically, we generate attacked datasets with NETTACK [71] with respect to different perturbation number $n$ in both evasive attacking mode and poisoning attacking mode. Higher $n$ represents a heavier attacking degree. Results are reported in Table 2.

**Results**. Similar trends can be found that DGIB-Bern and DGIB-Cat outperform all baselines under different settings. DGIB-Bern improves the AUC over the average baselines by 8.4%, 5.4%, and 27.8% under evasive attacks for three datasets, while DGIB-Cat surpasses 6.4%, 3.9%, and 13.4%. In poisoning attack settings, DGIB-Cat

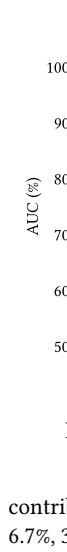

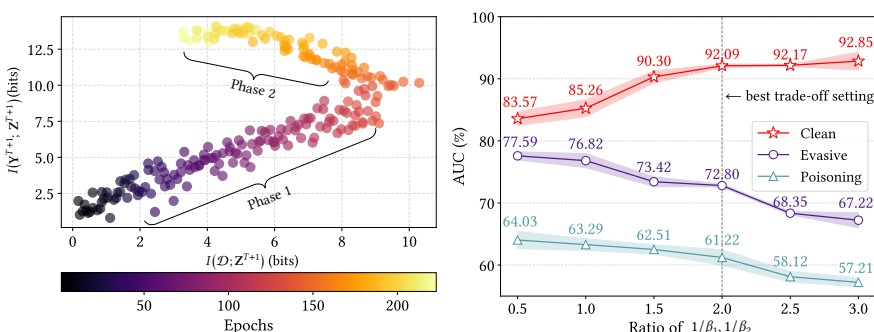

**Figure 4: Ablation study results.**  **Figure 5: Information Plane analysis.**  **Figure 6: Performance trade-off analysis.**

contributes 8.0%, 4.6%, and 13.4% increase, and DGIB-Bern improves 6.7%, 3.5%, and 13.4%. Results demonstrate DGIB is contained under challenging evasive and poisoning adversarial attacks.

Comprehensively, DGIB-Bern owns a better performance in targeted adversarial attacks with a lower average AUC decrease, while DGIB-Bern is better in non-targeted adversarial attacks. We explain this phenomenon as the $\text{Cat}(\cdot)$ is the non-informative distribution, which fits well to the non-targeted settings, and $\text{Bern}(\cdot)$ requires priors, which may be more appropriate for against targeted attacks.

### 5.4 Ablation Study

In this section, we analyze the effectiveness of the three variants:

- **DGIB ($w/o$ *Cons*)**: We remove the *Consensual* channel $\text{DGIB}_C$ in the overall training objective (Eq. (22)), and optimizing only with the *Minimal* and *Sufficient* channel $\text{DGIB}_{MS}$.
- **DGIB ($w/o$ $\mathcal{A}$)**: We remove the structure sampling term ($\mathcal{A}$) in the upper bound of $I(\mathcal{D}; \mathbf{Z}^{T+1})$ (Eq. (11)).
- **DGIB ($w/o$ $\mathcal{Z}$)**: We remove the feature sampling term ($\mathcal{Z}$) in the upper bound of $I(\mathcal{D}; \mathbf{Z}^{T+1})$ (Eq. (11)).

We choose DGIB-Bern as the backbone and compare performances on the clean, evasive attacked ($n = 2$) and poisoning attacked ($n = 2$) COLLAB, respectively. Results are shown in Figure 4.

**Results.** Overall, DGIB outperforms the other three variants, except for DGIB ($w/o$ $\mathcal{A}$), where it exceeds the original DGIB-Bern on the clean COLLAB by 1.1%. We claim this phenomenon is within our expectation as the structure sampling term ($\mathcal{A}$) contributes to raising the robustness by refining structures and compression feature information, which will surely damage its performance on the clean dataset. Concretely, we witness DGIB surpassing all three variants when confronting evasive and poisoning adversarial attacks, which provides insights into the effectiveness of the proposed components and demonstrates their importance in achieving better performance for robust representation learning on dynamic graphs.

### 5.5 Information Plane Analysis

In this section, we observe the evolution of the IB compression process on the Information Plane, which is widely applied to analyze the changes in the mutual information between input, latent representations, and output during training. Given the Markov Chain $< \mathbf{X} \rightarrow \mathbf{Y} \rightarrow \mathbf{Z} >$, the latent representation is uniquely mapped to

a point in the Information Plane with coordinates $\big(I(\mathbf{X}; \mathbf{Z}), I(\mathbf{Y}, \mathbf{Z})\big)$. We analyze DGIB-Bern on the clean COLLAB and draw the coordinates of $\big(I(\mathcal{D}; \mathbf{Z}^{T+1}), I(\mathbf{Y}^{T+1}; \mathbf{Z}^{T+1})\big)$ in Figure 5.

**Results.** We note that the information evolution process is composed of two phases. During Phase 1 (ERM phase), $I(\mathcal{D}; \mathbf{Z}^{T+1})$ and $I(\mathbf{Y}^{T+1}; \mathbf{Z}^{T+1})$ both increase, indicating the latent representations are extracting information about the input and labels, and the coordinates on the Information Plane are shifting toward the upper right corner. In Phase 2 (compression phase), $I(\mathcal{D}; \mathbf{Z}^{T+1})$ begins to decline, while the growth rate of $I(\mathbf{Y}^{T+1}; \mathbf{Z}^{T+1})$ slows down, and converging to the upper left corner, indicating our DGIB principle begins to take effect, leading to a *Minimal* and *Sufficient* latent representation with *Consensual* Constraint.

### 5.6 Hyperparameter Trade-off Analysis

In this section, we analyze the impact of the compression parameters $\beta_1$ and $\beta_2$ on the trade-off between the performance of prediction and robustness. We conduct experiments based on DGIB-Bern with different ratios of MSE term and compression term $(1/\beta_1, 1/\beta_2)$ on the clean, evasive attacked ($n = 2$) and poisoning attacked ($n = 2$) COLLAB, respectively. Results are reported in Figure 6.

**Results.** With the increase of $1/\beta_1$ or $1/\beta_2$, DGIB-Bern performs better on the clean COLLAB, while its robustness against targeted adversarial attacks decreases. This validates the conflicts between the MSE term and the compression term, as the compression term sacrifices prediction performance to improve the robustness. We find the optimal setting for $1/\beta_1$ and $1/\beta_2$, where both the performance of prediction and robustness are well-preserved.

## 6 CONCLUSION

In this paper, we present a novel framework named **DGIB** to learn robust and discriminative representations on dynamic graphs grounded in the information-theoretic IB principle for the first time. We decompose the overall DGIB objectives into $\text{DGIB}_{MS}$ and $\text{DGIB}_C$ channels, which act jointly to satisfy the proposed *MSC* Condition that the optimal representations should satisfy. Variational bounds are further introduced to efficiently and appropriately estimate intractable IB terms. Extensive experiments demonstrate the superior robustness of DGIB against adversarial attacks compared with state-of-the-art baselines in the link prediction task.

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

# A DETAILED OPTIMIZATION AND COMPLEXITY ANALYSIS

We illustrate the overall training pipeline of DGIB in Algorithm 1.

---

**Algorithm 1:** Overall training pipeline of DGIB framework.

**Input:** Dynamic graph $\mathbf{DG} = \{\mathcal{G}^t\}_{t=1}^{T}$; Node features $\mathbf{X}^{1:T+1}$; Labels $\mathbf{Y}^{1:T}$ of link occurrence; Number of layers $L$; Relative time encoding function $\text{RTE}(\cdot)$; Non-linear rectifier $\tau$; Activation function $\sigma$; Hyperparameters $k$, $\alpha$, $\beta_1$ and $\beta_2$.

**Output:** The optimized robust model $f_{\boldsymbol{\theta}}^{\star} = w \circ g$; Predicted label $\hat{\mathbf{Y}}^{T+1}$ of link occurrence at time $T + 1$.

1 Initialize weights $\mathbf{W}$ and learnable parameters $\boldsymbol{\theta}$ randomly;

2 $\mathbf{Z}^{1:T+1,(0)} \leftarrow \text{RTE}(\mathbf{X}^{1:T+1})$, attentions $\mathbf{w} \leftarrow \text{GAT}(\mathbf{Z}^{1:T+1,(0)})$;

3 Construct $\mathcal{N}_{ST}(v, k, t) \leftarrow \{u \in \mathcal{V}^{t-1:t} \mid d(u, v) = k\}$;

4 **for** $l = 1, 2, \cdots, L$ and $v \in \mathcal{V}^{1:T+1}$ **do**

5    **for** $t$ in range $[T + 1]$ **do**

6      $\hat{\mathbf{Z}}^{t(l-1)} \leftarrow \tau(\mathbf{Z}^{t(l-1)})\mathbf{W}^{(l)}$;

7      $\phi_{v,k}^{t(l)} \leftarrow \sigma\{(\hat{\mathbf{Z}}_v^{t(l-1)} \| \hat{\mathbf{Z}}_u^{t-1:t(l-1)})\mathbf{w}^\top\}_{u \in \mathcal{N}_{ST}(v,k,t)}$;

8      $\hat{\mathbf{A}}^{t(l)} \leftarrow \cup_{v \in \mathcal{V}^t} \{u \in \mathcal{N}_{ST}(v, k, t) | u \sim \text{Bern}(\phi_{v,k}^{t(l)})\}$

9      or $\cup_{v \in \mathcal{V}^t} \{u \in \mathcal{N}_{ST}(v, k, t) | u \sim \text{Cat}(\phi_{v,k}^{t(l)})\}$;

10      $\mathbf{Z}^{t(l)} \leftarrow \sum_{(u,v) \in \hat{\mathbf{A}}^{t(l)}} \{\hat{\mathbf{Z}}_v^{t(l-1)}\}_{v \in \mathcal{V}^t}$;

11    **end**

12 $\hat{\mathbf{Y}}^{T+1} \leftarrow g(\mathbf{Z}^{T+1(L)})$;

13 $\text{DGIB}_{MS} \leftarrow$ Eq. (9), $\text{DGIB}_C \leftarrow$ Eq. (14);

14 Calculate the overall loss, as $\mathcal{L}_{\text{DGIB}} \leftarrow$ Eq. (22);

15 Update $\boldsymbol{\theta}$ by minimizing $\mathcal{L}_{\text{DGIB}}$ and back-propagation.

16 **end**

---

**Comlexity Analysis.** We analyze the computational complexity of each part in DGIB as follows. For brevity, denote $|\mathcal{V}|$ and $|\mathcal{E}|$ as the total number of nodes and edges in each graph snapshot, respectively. Let $d$ be the dimension of the input node features, and $d'$ be the dimension of the latent node representation.

- Linear input feature projection layer: $O((T + 1)|\mathcal{V}|dd')$.
- Relative Time Encoding ($\text{RTE}(\cdot)$) layer: $O((T + 1)|\mathcal{V}|d'^2)$.
- Spatio-temporal neighbor sampling and attention computation: $O(L(T+1)|\mathcal{V}|) + O(2Lk(T+1)|\mathcal{V}||\mathcal{E}|d')$, where $k$ is the range of receptive field, and $L$ is the number of layers.
- Feature aggregation: Constant complexity brought about by addition operations (ignored).
- Link prediction: $O(|\mathcal{E}'|d')$, where $\mathcal{E}'$ is the number of sampled links to be predicted.

In summary, the overall computational complexity of DGIB is:

$$O(2Lk(T+1)|\mathcal{V}||\mathcal{E}|dd'^2) + O(|\mathcal{E}'|d'). \tag{A.1}$$

In conclusion, DGIB has a *linear* computation complexity with respect to the number of nodes and edges in all graph snapshots, which is on par with the state-of-the-art DGNNs. In addition, based on our experiments experience, DGIB can be trained and tested under the hardware configurations (including memory requirements) listed in Appendix D.3, and the training time consumption is listed in Appendix C.5, which demonstrates our DGIB has similar time complexity compared with most of the existing DGNNs.

## B  PROOFS

In this section, we provide proofs of Proposition 1, Proposition 2, and Proposition 3.

### B.1  Proof of Proposition 1

We restate Proposition 1:

**Proposition 1** (The Lower Bound of $I(\mathbf{Y}^{T+1}; \mathbf{Z}^{T+1})$). For any distributions $\mathbb{Q}_1(\mathbf{Y}^{T+1} \mid \mathbf{Z}^{T+1})$ and $\mathbb{Q}_2(\mathbf{Y}^{T+1})$:

$$
I(\mathbf{Y}^{T+1}; \mathbf{Z}^{T+1}) \geq 1 + \mathbb{E}_{\mathbb{P}(\mathbf{Y}^{T+1}, \mathbf{Z}^{T+1})} \left[ \log \frac{\mathbb{Q}_1(\mathbf{Y}^{T+1} \mid \mathbf{Z}^{T+1})}{\mathbb{Q}_2(\mathbf{Y}^{T+1})} \right]
$$
$$
- \mathbb{E}_{\mathbb{P}(\mathbf{Y}^{T+1})\mathbb{P}(\mathbf{Z}^{T+1})} \left[ \frac{\mathbb{Q}_1(\mathbf{Y}^{T+1} \mid \mathbf{Z}^{T+1})}{\mathbb{Q}_2(\mathbf{Y}^{T+1})} \right]. \quad (B.1)
$$

PROOF. We apply the variational bounds of mutual information $I_{\text{NWJ}}$ proposed by [39], which is thoroughly concluded in [42].

**Lemma 1** (Mutual Information Variational Bounds in $I_{\text{NWJ}}$). Given any two variables $\mathbf{X}$ and $\mathbf{Y}$, and any permutation invariant function $f(\mathbf{X}, \mathbf{Y})$, we have:

$$
I(\mathbf{X}; \mathbf{Y}) \geq \mathbb{E}_{\mathbb{P}(\mathbf{X}, \mathbf{Y})} \left[ f(\mathbf{X}, \mathbf{Y}) \right] - e^{-1} \mathbb{E}_{\mathbb{P}(\mathbf{X})\mathbb{P}(\mathbf{Y})} \left[ e^{f(\mathbf{X}, \mathbf{Y})} \right]. \quad (B.2)
$$

As $f(\mathbf{X}, \mathbf{Y})$ must learn to self-normalize, yielding a unique solution for variables $(\mathbf{Y}^{T+1}, \mathbf{Z}^{T+1})$ by plugging:

$$
f(\mathbf{Y}^{T+1}, \mathbf{Z}^{T+1}) = 1 + \log \frac{\mathbb{Q}_1(\mathbf{Y}^{T+1} \mid \mathbf{Z}^{T+1})}{\mathbb{Q}_2(\mathbf{Y}^{T+1})} \quad (B.3)
$$

into Eq. (B.2). Specifically:

$$
I(\mathbf{Y}^{T+1}; \mathbf{Z}^{T+1}) \geq \mathbb{E}_{\mathbb{P}(\mathbf{Y}^{T+1}, \mathbf{Z}^{T+1})} \left[ 1 + \log \frac{\mathbb{Q}_1(\mathbf{Y}^{T+1} \mid \mathbf{Z}^{T+1})}{\mathbb{Q}_2(\mathbf{Y}^{T+1})} \right]
$$
$$
- e^{-1} \mathbb{E}_{\mathbb{P}(\mathbf{X})\mathbb{P}(\mathbf{Y})} \left[ e \cdot \frac{\mathbb{Q}_1(\mathbf{Y}^{T+1} \mid \mathbf{Z}^{T+1})}{\mathbb{Q}_2(\mathbf{Y}^{T+1})} \right] \quad (B.4)
$$
$$
\geq 1 + \mathbb{E}_{\mathbb{P}(\mathbf{Y}^{T+1}, \mathbf{Z}^{T+1})} \left[ \log \frac{\mathbb{Q}_1(\mathbf{Y}^{T+1} \mid \mathbf{Z}^{T+1})}{\mathbb{Q}_2(\mathbf{Y}^{T+1})} \right]
$$
$$
- \mathbb{E}_{\mathbb{P}(\mathbf{Y}^{T+1})\mathbb{P}(\mathbf{Z}^{T+1})} \left[ \frac{\mathbb{Q}_1(\mathbf{Y}^{T+1} \mid \mathbf{Z}^{T+1})}{\mathbb{Q}_2(\mathbf{Y}^{T+1})} \right]. \quad (B.5)
$$

We conclude the proof of Proposition 1. □

### B.2  Proof of Proposition 2

We restate Proposition 2:

**Proposition 2** (The Upper Bound of $I(\mathcal{D}; \mathbf{Z}^{T+1})$). Let $\mathcal{I}_{\mathbf{A}}, \mathcal{I}_{\mathbf{Z}} \subset [T+1]$ be the stochastic time indices sets. Based on the Markov property $\mathcal{D} \perp\!\!\!\perp \mathbf{Z}^{T+1} \mid (\{\hat{\mathbf{A}}^t\}_{t \in \mathcal{I}_{\mathbf{A}}} \cup \{\mathbf{Z}^t\}_{t \in \mathcal{I}_{\mathbf{Z}}})$, for any distributions

$\mathbb{Q}(\hat{\mathbf{A}}^t)$ and $\mathbb{Q}(\mathbf{Z}^t)$:

$$
I(\mathcal{D}; \mathbf{Z}^{T+1}) \leq I(\mathcal{D}; \{\hat{\mathbf{A}}^t\}_{t \in \mathcal{I}_{\mathbf{A}}} \cup \{\mathbf{Z}^t\}_{t \in \mathcal{I}_{\mathbf{Z}}}) \leq \sum_{t \in \mathcal{I}_{\mathbf{A}}} \mathcal{A}^t + \sum_{t \in \mathcal{I}_{\mathbf{Z}}} \mathcal{Z}^t,
$$
$$
(B.6)
$$

$$
where \quad \mathcal{A}^t = \mathbb{E} \left[ \log \frac{\mathbb{P}(\hat{\mathbf{A}}^t \mid \hat{\mathbf{Z}}^t, \mathbf{Z}^{t-1}, \mathbf{A}^t)}{\mathbb{Q}(\hat{\mathbf{A}}^t)} \right], \quad (B.7)
$$

$$
\mathcal{Z}^t = \mathbb{E} \left[ \log \frac{\mathbb{P}(\mathbf{Z}^t \mid \hat{\mathbf{Z}}^t, \mathbf{Z}^{t-1}, \hat{\mathbf{A}}^t)}{\mathbb{Q}(\mathbf{Z}^t)} \right]. \quad (B.8)
$$

PROOF. We apply the Data Processing Inequality (DPI) [5] and the Markovian dependency to prove the first inequality.

**Lemma 2** (Mutual Information Lower Bound in Markov Chain). Given any three variables $\mathbf{X}$, $\mathbf{Y}$ and $\mathbf{Z}$, which follow the Markov Chain $< \mathbf{X} \rightarrow \mathbf{Y} \rightarrow \mathbf{Z} >$, we have:

$$
I(\mathbf{X}; \mathbf{Y}) \geq I(\mathbf{X}; \mathbf{Z}). \quad (B.9)
$$

Directly applying Lemma 2 to the Markov Chain in DGIB, *i.e.*, $< \mathcal{D} \rightarrow (\{\hat{\mathbf{A}}^t\}_{t \in \mathcal{I}_{\mathbf{A}}} \cup \{\mathbf{Z}^t\}_{t \in \mathcal{I}_{\mathbf{Z}}}) \rightarrow \mathbf{Z}^{T+1} >$, which satisfies the Markov property $\mathcal{D} \perp\!\!\!\perp \mathbf{Z}^{T+1} \mid (\{\hat{\mathbf{A}}^t\}_{t \in \mathcal{I}_{\mathbf{A}}} \cup \{\mathbf{Z}^t\}_{t \in \mathcal{I}_{\mathbf{Z}}})$, we the have:

$$
I(\mathcal{D}; \mathbf{Z}^{T+1}) \leq I(\mathcal{D}; \{\hat{\mathbf{A}}^t\}_{t \in \mathcal{I}_{\mathbf{A}}} \cup \{\mathbf{Z}^t\}_{t \in \mathcal{I}_{\mathbf{Z}}}). \quad (B.10)
$$

Next, we prove the second inequality. To guarantee the compression order following "structure first, features second" in each time step, as well as the Spatio-Temporal Local Dependence (Assumption 2), we define the order:

**Definition 2** (DGIB Markovian Decision Order $\prec$). For any variables in set $\{\hat{\mathbf{A}}^t\}_{t \in \mathcal{I}_{\mathbf{A}}} \cup \{\mathbf{Z}^t\}_{t \in \mathcal{I}_{\mathbf{Z}}}$, we have:

- For different time indices $t_1$ and $t_2$, $\hat{\mathbf{A}}^{t_1}, \mathbf{Z}^{t_1} \prec \hat{\mathbf{A}}^{t_2}, \mathbf{Z}^{t_2}$.
- For any individual time index $t$, $\hat{\mathbf{A}}^t \prec \mathbf{Z}^t$.

To satisfy the Spatio-Temporal Local Dependence Assumption, we define two preceding sequences of sets based on the $\prec$ order:

$$
\mathcal{S}_{\hat{\mathbf{A}}}^t, \mathcal{S}_{\mathbf{Z}}^t = \{\hat{\mathbf{A}}^{t_1}, \mathbf{Z}^{t_1} \mid t_1, t_2 < t - 1, t_1 \in \mathcal{I}_{\mathbf{A}}, t_2 \in \mathcal{I}_{\mathbf{Z}}\}. \quad (B.11)
$$

Thus, we have $\mathcal{S}_{\hat{\mathbf{A}}}^t \perp\!\!\!\perp \hat{\mathbf{A}}^t \mid \{\mathbf{Z}^{t-1}, \mathbf{A}^t\}$ and $\mathcal{S}_{\mathbf{Z}}^t \perp\!\!\!\perp \mathbf{Z}^t \mid \{\mathbf{Z}^{t-1}, \hat{\mathbf{A}}^t\}$. Then, we decompose $I(\mathcal{D}; \{\hat{\mathbf{A}}^t\}_{t \in \mathcal{I}_{\mathbf{A}}} \cup \{\mathbf{Z}^t\}_{t \in \mathcal{I}_{\mathbf{Z}}})$ into:

$$
I(\mathcal{D}; \{\hat{\mathbf{A}}^t\}_{t \in \mathcal{I}_{\mathbf{A}}} \cup \{\mathbf{Z}^t\}_{t \in \mathcal{I}_{\mathbf{Z}}}) = \sum_{t \in \mathcal{I}_{\mathbf{A}}} I(\mathcal{D}; \hat{\mathbf{A}}^t \mid \mathcal{S}_{\hat{\mathbf{A}}}^t)
$$
$$
+ \sum_{t \in \mathcal{I}_{\mathbf{Z}}} I(\mathcal{D}; \mathbf{Z}^t \mid \mathcal{S}_{\mathbf{Z}}^t). \quad (B.12)
$$

Following [58], we provide upper bounds for $I(\mathcal{D}; \hat{\mathbf{A}}^t \mid \mathcal{S}_{\hat{\mathbf{A}}}^t)$ and $I(\mathcal{D}; \mathbf{Z}^t \mid \mathcal{S}_{\mathbf{Z}}^t)$, respectively:

$$
I(\mathcal{D}; \hat{\mathbf{A}}^t \mid \mathcal{S}_{\hat{\mathbf{A}}}^t) \leq I(\mathcal{D}, \mathbf{Z}^{t-1}; \hat{\mathbf{A}}^t \mid \mathcal{S}_{\hat{\mathbf{A}}}^t) = I(\mathbf{Z}^{t-1}, \mathbf{A}^t; \hat{\mathbf{A}}^t \mid \mathcal{S}_{\hat{\mathbf{A}}}^t)
$$
$$
\leq I(\mathbf{Z}^{t-1}, \mathbf{A}^t; \hat{\mathbf{A}}^t) = \mathcal{A}^t - \text{KL} \left[ \mathbb{P}(\hat{\mathbf{A}}^t) \| \mathbb{Q}(\hat{\mathbf{A}}^t) \right]
$$
$$
\leq \mathcal{A}^t. \quad (B.13)
$$

Similarly, we have:

$$
I(\mathcal{D}; \mathbf{Z}^t \mid \mathcal{S}_{\mathbf{Z}}^t) \leq I(\mathcal{D}, \mathbf{Z}^{t-1}, \hat{\mathbf{A}}^t; \mathbf{Z}^t \mid \mathcal{S}_{\mathbf{Z}}^t) = I(\mathbf{Z}^{t-1}, \hat{\mathbf{A}}^t; \mathbf{Z}^t \mid \mathcal{S}_{\mathbf{Z}}^t)
$$
$$
\leq I(\mathbf{Z}^{t-1}, \hat{\mathbf{A}}^t; \mathbf{Z}^t) = \mathcal{Z}^t - \text{KL} \left[ \mathbb{P}(\mathbf{Z}^t) \| \mathbb{Q}(\mathbf{Z}^t) \right]
$$
$$
\leq \mathcal{Z}^t. \quad (B.14)
$$

By plugging Eq. (B.13) and Eq. (B.14) into Eq. (B.12), we conclude the proof of the second inequality. □

## B.3 Proof of Proposition 3

We restate Proposition 3:

**Proposition 3** (The Upper Bound of $I(\mathbf{Z}^{1:T}; \mathbf{Z}^{T+1})$). For any distributions $\mathbb{Q}(\mathbf{Z}^{T+1})$:

$$I(\mathbf{Z}^{1:T}; \mathbf{Z}^{T+1}) \leq \mathbb{E}\left[\log \frac{\mathbb{P}(\mathbf{Z}^{T+1} \mid \mathbf{Z}^{1:T})}{\mathbb{Q}(\mathbf{Z}^{T+1})}\right]. \tag{B.15}$$

Proof. We apply the upper bound proposed in the Variational Information Bottleneck (VIB) [1].

**Lemma 3** (Mutual Information Upper Bound in VIB). Given any two variables $\mathbf{X}$ and $\mathbf{Y}$, we have the variational upper bound of $I(\mathbf{X}; \mathbf{Y})$:

$$I(\mathbf{X}; \mathbf{Y}) = \mathbb{E}_{\mathbb{P}(\mathbf{X}, \mathbf{Y})}\left[\log \frac{\mathbb{P}(\mathbf{Y} \mid \mathbf{X})}{\mathbb{P}(\mathbf{Y})}\right] = \mathbb{E}_{\mathbb{P}(\mathbf{X}, \mathbf{Y})}\left[\log \frac{\mathbb{P}(\mathbf{Y} \mid \mathbf{X})\mathbb{Q}(\mathbf{Y})}{\mathbb{P}(\mathbf{Y})\mathbb{Q}(\mathbf{Y})}\right]$$

$$= \mathbb{E}_{\mathbb{P}(\mathbf{X}, \mathbf{Y})}\left[\log \frac{\mathbb{P}(\mathbf{Y} \mid \mathbf{X})}{\mathbb{Q}(\mathbf{Y})}\right] - \underbrace{\text{KL}\left[\mathbb{P}(\mathbf{Y})\|\mathbb{Q}(\mathbf{Y})\right]}_{\text{non-negative}}$$

$$\leq \mathbb{E}_{\mathbb{P}(\mathbf{X}, \mathbf{Y})}\left[\log \frac{\mathbb{P}(\mathbf{Y} \mid \mathbf{X})}{\mathbb{Q}(\mathbf{Y})}\right]. \tag{B.16}$$

By utilizing Lemma 3 to variables $\mathbf{Z}^{1:T}$ and $\mathbf{Z}^{T+1}$, we conclude the proof of Proposition 3.

□

## C EXPERIMENT DETAILS AND ADDITIONAL RESULTS

In this section, we provide additional experiment details and results.

### C.1 Datasets Details

We use three real-world datasets to evaluate DGIB on the challenging future link prediction task.

- **COLLAB**[1] [53] is an academic collaboration dataset with papers that were published during 1990-2006 (16 graph snapshots), which reveals the dynamic citation networks among authors. Nodes and edges represent authors and co-authorship, respectively. Based on the co-authored publication, there are five attributes in edges, including "Data Mining", "Database", "Medical Informatics", "Theory" and "Visualization". We apply word2vec [35] to extract 32-dimensional node features from paper abstracts. We use 10/1/5 chronological graph snapshots for training, validation, and testing, respectively. The dataset includes 23,035 nodes and 151,790 links in total.
- **Yelp**[2] [46] contains customer reviews on business, which are collected from the crowd-sourced local business review and social networking web. Nodes represent customer or business, and edges represent review behaviors, respectively. Considering categories of business, there are five attributes in edges, including "Pizza", "American (New) Food", "Coffee & Tea", "Sushi Bars" and "Fast Food" from January 2019 to December 2020 (24 graph

snapshots). We apply word2vec [35] to extract 32-dimensional node features from reviews. We use 15/1/8 chronological graph snapshots for training, validation, and testing, respectively. The dataset includes 13,095 nodes and 65,375 links in total.
- **ACT**[3] [29] describes student actions on a popular MOOC website within a month (30 graph snapshots). Nodes represent students or targets of actions, edges represent actions. Considering the attributes of different actions, we apply K-Means to cluster the action features into five categories. We assign the features of actions to each student or target and expand the original 4-dimensional features to 32 dimensions by a linear function. We use 20/2/8 chronological graph snapshots for training, validation, and testing, respectively. The dataset includes 20,408 nodes and 202,339 links in total.

Statistics of the three datasets are concluded in Table B.1. These three datasets have different time spans and temporal granularity (16 years, 24 months, and 30 days), covering most real-world scenarios. The most challenging dataset for the future link prediction task is the COLLAB. In addition to having the longest time span and the coarsest temporal granularity, it also has the largest difference in the properties of its links, which greatly challenges the robustness.

**Table B.1: Statistics of the real-world dynamic graph datasets.**

| Dataset | # Node | # Link | # Link Type | Length (Split) | Temporal Granularity |
|---|---|---|---|---|---|
| COLLAB | 23,035 | 151,790 | 5 | 16 (10/1/5) | year |
| Yelp | 13,095 | 65,375 | 5 | 24 (15/1/8) | month |
| ACT | 20,408 | 202,339 | 5 | 30 (20/2/8) | day |

### C.2 Detailed Settings and Analysis for Experiment in Figure 3

In Figure 3, we evaluate the performance of our DGIBagainst non-targeted adversarial attack compared with GIB [58]. We choose the ACT [29] as the dataset, with 20 graphs to train, 2 graphs to validate, and 8 graphs to test. In attacking settings, we follow the same non-targeted adversarial attack settings introduced in Section 5.1.3 on graph structures. For the baseline, we adapt the GIB [58]-Cat to dynamic scenarios by obtaining $\hat{\mathbf{Z}}^t$ for each individual graph snapshot first with the original GIB [58], and then aggregating them using the vanilla LSTM [19] to learn the comprehensive representation $\hat{\mathbf{Z}}^{T+1}$, where the two steps are jointly optimized by the overall objective Eq. (4). For our DGIB, we train the DGIB-Cat version with the same objective in Eq. (4).

Results show that, for GIB (adapted), we witness a sudden drop in the link prediction performance (AUC %) achieved by $\hat{\mathbf{Z}}^{t+1}$ for the next time-step graph $\mathcal{G}^{t+1}$ in testing, while our DGIB-Cat contains a slight and acceptable decrease when encountering adversarial attacks, and its average testing score surpasses GIB. This demonstrates that directly optimizing GIB with the intuitive IB objective in Eq. (4) will lead to sub-optimal model performance, and our DGIB is endowed with stronger robustness by jointly optimizing $\text{DGIB}_{MS}$ and $\text{DGIB}_C$ channels, which cooperate and constrain each other to satisfy the *MSC* Condition.

---

[1] https://www.aminer.cn/collaboration
[2] https://www.yelp.com/dataset

[3] https://snap.stanford.edu/data/act-mooc.html

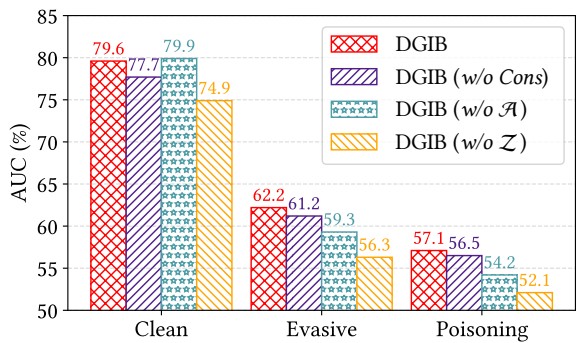

Figure B.1: Results of ablation study on Yelp.

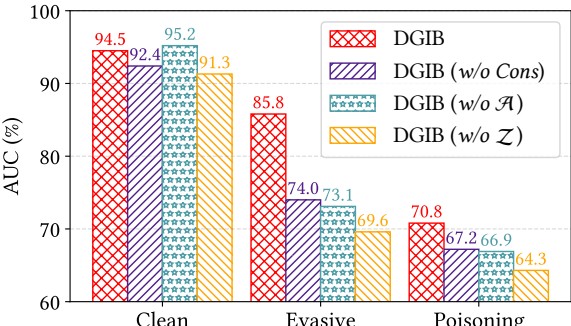

Figure B.2: Results of ablation study on ACT.

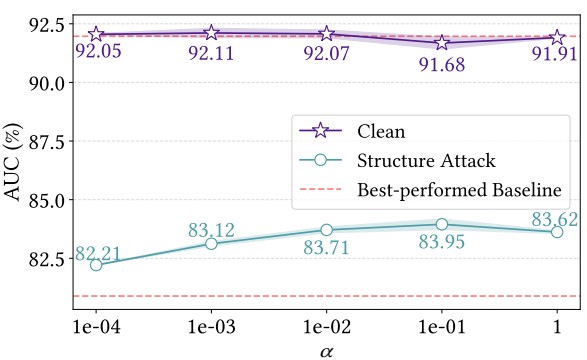

Figure B.3: Sensitivity analysis of $\alpha$ on COLLAB.

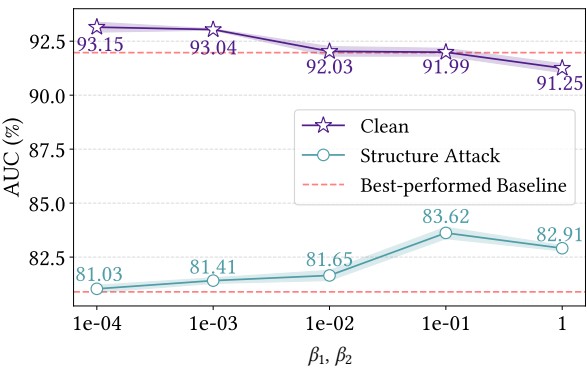

Figure B.4: Sensitivity analysis of $\beta_1, \beta_2$ on COLLAB.

## C.3 Additional Results of Ablation Study

We provide additional results of ablation studies on dynamic graph datasets Yelp and ACT. Accordingly, we choose DGIB-Bern as the backbone and compare performances on the clean, evasive attacked ($n = 2$) and poisoning attacked ($n = 2$) Yelp and ACT, respectively. Results are shown in Figure B.1 and Figure B.2.

**Results.** Similar conclusions can be derived as in Section 5.4 that DGIB outperforms the other three variants, except for DGIB (*w/o* $\mathcal{A}$), where it exceeds the original DGIB-Bern on the clean Yelp and clean ACT by 0.3% and 0.7%, respectively. We explain this phenomenon that the structure sampling term ($\mathcal{A}$) acts to improve the robustness by modifying structures and compression features, which will damage the performance on clean occasions. When confronting evasive and poisoning adversarial attacks, DGIB surpassing all three variants, which validates the importance of the proposed three mechanisms in achieving better performance for robust representation learning on dynamic graphs.

## C.4 Parameter Sensitivity Analysis

We provide additional experiments on the sensitivity of hyperparameters $\alpha$, $\beta_1$ and $\beta_2$, which are chosen from {1e-04, 1e-03, 1e-02, 1e-01, 1}. We report the results of sensitivity analysis on the clean and structure-attacked COLLAB in Figure B.3 and Figure B.4.

**Results.** Results demonstrate that the performance on both the clean and structure-attacked COLLAB is sensitive to different

values of $\alpha$, $\beta_1$, and $\beta_2$, and contains in a reasonable range. In addition, we observe there exist negative correlations between performance on the clean dataset and attacked dataset for three hyperparameters, which demonstrates the confrontations of the MSE term and compression term in DGIB overall optimization objectives (Eq. (7)). Specifically, in most cases, higher $\alpha$, $\beta_1$, and $\beta_2$, lead to better robustness but weaker clean dataset performance. In conclusion, different combinations of hyperparameters lead to varying task performance and model robustness, and we follow the tradition of configuring the values of hyperparameters with the best trade-off setting against adversarial attacks.

## C.5 Training Efficiency Analysis

We report the training time for our DGIB-Bern and DGIB-Cat with the default configurations as we provided in the code. We conduct experiments with the hardware and software configurations listed in Section D.3. We ignore the tiny difference when training under the poisoning attacks and only report the average time per epoch for training on the respective clean datasets in Table B.2.

**Results.** The training time of DGIB-Bern and DGIB-Cat is in the same order as the state-of-the-art DGNN baselines due to the computation complexity of DGIB is on par with related works. The maximum epochs for each training is set to 1000 (fixed), thus the total time is feasible in practice. Note that, the number of layers, neighbors to sampling, *etc.*, will have a significant impact on the

training efficiency, and not always large numbers bring extra improvements in performance, so it is recommended to properly set the related parameters.

**Table B.2: Results of training time per epoch (s).**

| Dataset | COLLAB | Yelp | ACT |
|---------|--------|------|-----|
| DIDA [67] | 2.61 | 5.12 | 3.31 |
| DGIB-Bern | 0.88 | 1.91 | 1.32 |
| DGIB-Cat | 0.86 | 1.54 | 1.64 |

## D  IMPLEMENTATION DETAILS

In this section, we provide implementation details of the proposed DGIB and baselines.

### D.1  DGIB Implementation Details

According to respective experiment settings, we randomly split the dynamic graph datasets into training, validation, and testing chronological sets. We sample negative links from nodes that do not have links, and the negative links for validation and testing sets are kept the same for all baseline methods and ours. We set the number of positive links to the same as the negative links. We use the Area under the ROC Curve (AUC) as the evaluation metric. As we focus on the future link prediction task, we use the inner product of a pair of learned node representations to predict the occurrence of links, *i.e.*, we implement the link predictor $g(\cdot)$ as the inner product of hidden embeddings, which is widely applied in future link prediction tasks. The non-linear rectifier $\tau$ is ReLU$(\cdot)$ [37], and the activation function $\sigma$ is Sigmoid$(\cdot)$ [17]. We randomly run all the experiments for five times, and report the average results with standard deviations.

### D.2  Baseline Implementation Details

We provide the baseline methods implementations with respective licenses as follows.

- **GAE** [27]: https://github.com/DaehanKim/vgae_pytorch.
- **VGAE** [27]: https://github.com/DaehanKim/vgae_pytorch.
- **GAT** [56]: https://github.com/pyg-team/pytorch_geometric.
- **GCRN** [48]: https://github.com/youngjoo-epfl/gconvRNN.
- **EvolveGCN** [41]: https://github.com/IBM/EvolveGCN.
- **DySAT** [46]: https://github.com/FeiGSSS/DySAT_pytorch.
- **IRM** [2]: https://github.com/facebookresearch/InvariantRiskMinimization.
- **V-REx** [28]: https://github.com/capybaralet/REx_code_release.
- **GroupDRO** [45]: https://github.com/kohpangwei/group_DRO.
- **RGCN** [69]: https://github.com/DSE-MSU/DeepRobust.
- **DIDA** [67]: https://github.com/wondergo2017/DIDA.
- **GIB** [58]: https://github.com/snap-stanford/GIB. We report the best result between GIB-Bern and GIB-Cat versions.
- **NETTACK** [71]: https://github.com/DSE-MSU/DeepRobust.

The parameters of baseline methods are set as the suggested value in their papers or carefully tuned for fairness.

## D.3  Hardware and Software Configurations

We conduct the experiments with:

- Operating System: Ubuntu 20.04 LTS.
- CPU: Intel(R) Xeon(R) Platinum 8358 CPU@2.60GHz with 1TB DDR4 of Memory.
- GPU: NVIDIA Tesla A100 SMX4 with 40GB of Memory.
- Software: CUDA 10.1, Python 3.8.12, PyTorch[4] 1.9.1, PyTorch Geometric[5] 2.0.1.

---

[4] https://github.com/pytorch/pytorch
[5] https://github.com/pyg-team/pytorch_geometric

