# OpenReview forum: "Dynamic Graph Information Bottleneck"
_ACM.org/TheWebConf/2024/Conference — TheWebConf24 Oral_

### Official Review · Reviewer_GgeH · 2023-11-10

**Novelty:** 6
**Technical Quality:** 6

**Review:**

This paper presented a novel dynamic graph information bottleneck (DGIB) framework for learning robust and discriminative representations from dynamic graphs. It introduced a Minimal-Sufficient-Consensual condition to guide the design of DGIB. Specially, DGIB considered the objective functions by learning the minimal and sufficient representations and enforcing the predictive consensus. Overall, the quality, clarity, originality, and significance of this work are given below.

**Quality:** The proposed DGIB approach was well motivated. The introduced Minimal-Sufficient-Consensual condition provided the guidelines to extend information bottleneck (IB) to dynamic graphs. In the experiments, the effectiveness of DGIB was also empirically verified in the link prediction task on real-world datasets.

**Clarity:** This paper started with the Minimal-Sufficient-Consensual (MSC) Condition when studying the dynamic graphs. Then this condition motivated the DGIB principle with two components DGIB$_{MS}$ and DGIB$_C$. Then the instantiation of DGIB was provided, followed by its computational complexity. The overall structure of this paper was easy to follow.

**Originality:** This paper focused on extending the IB to dynamic graphs, in order to learn robust and discriminative representations. The main technical novelties were (1) the Minimal-Sufficient-Consensual condition in the context of dynamic graph learning, and (2) the instantiated DGIB$_{MS}$ and DGIB$_C$ for efficient model training. Experimental results on target/non-target adversarial attacks demonstrated that the proposed DGIB method learned robust features for link prediction tasks.

**Significance:** It analyzed the information bottleneck in dynamic graphs and the proposed DGIB method achieved better performance over state-of-the-art baselines on several dynamic graphs.


Pros:

(1) This paper presented the Minimal-Sufficient-Consensual condition and showed how DGIB$_{MS}$ and DGIB$_C$ channels can refine the spatio-temporal information flow for representation learning.

(2) The variational bounds of DGIB$_{MS}$ and DGIB$_C$ were derived to make training objectives of DGIB tractable.

(3) Experiments demonstrated the robustness of DGIB against targeted and non-targeted adversarial attacks.

Cons:

(1) It might be better to show the training procedures of DGIB in section 4 (e.g., moving Algorithm 1 to the main body). Section 4.3 provides the overall objective functions. This makes the complexity analysis in section 4.4 hard to follow.

(2) The impact of the number of layers is unclear. In the hyperparamter setting, it sets the number of layers as one for DGIB to avoid overfitting. But such overfitting issues are not well explained in the experiments.

**Questions:**

(1) In line 289, $H(\cdot)$ denotes the information entropy. But $H(\cdot)$ is not used in previous formula.

(2) "$\mathbf{T}$" in line 316 is undefined.

**Reviewer Confidence:**

2: The reviewer is willing to defend the evaluation, but it is likely that the reviewer did not understand parts of the paper

**Scope:**

3: The work is somewhat relevant to the Web and to the track, and is of narrow interest to a sub-community

---

### Official Review · Reviewer_4UzM · 2023-11-19

**Novelty:** 6
**Technical Quality:** 6

**Review:**

This paper introduces the Dynamic Graph Information Bottleneck (DGIB) framework, a novel approach for learning robust and discriminative representations in dynamic graphs. The framework employs the Information Bottleneck (IB) principle and introduces the Minimal-Sufficient-Consensual (MSC) Condition to guide representation learning. DGIB operates by directing and refining the information flow through graph snapshots, decomposing its objectives into two channels: DGIB𝑀𝑆, focusing on minimal and sufficient representations, and DGIB𝐶, ensuring predictive consensus. The paper demonstrates the framework's robustness against adversarial attacks through extensive experiments on real-world and synthetic dynamic graph datasets, establishing DGIB as a pioneering approach in utilizing the IB principle for dynamic graphs.

Pros:
1. DGIB is the first framework to apply the Information Bottleneck principle in dynamic graph representation, marking a significant advancement in the field.
2. The framework demonstrates superior resistance to adversarial attacks, particularly in link prediction tasks, compared to existing state-of-the-art methods.
3. The introduction of the MSC Condition provides a new, theoretically sound basis for evaluating and guiding dynamic graph representation learning.

Cons:
1. The decomposition of IB objectives into two channels might add complexity, potentially impacting the interpretability of the framework.

2. The paper predominantly focuses on robustness against adversarial attacks, which, while important, may limit the exploration of other aspects of dynamic graph representation learning.

3. There is a lack of a reasonable criterion to determine whether to use DGIB-Bern or GIB-Cat.

**Questions:**

Could you elaborate on the specific mechanisms through which DGIB counters adversarial attacks more effectively compared to other methods?

The MSC Condition is a central aspect of your framework. How does it fundamentally differ from existing conditions or principles in dynamic graph representation learning?

**Ethics Review Description:**

None.

**Reviewer Confidence:**

2: The reviewer is willing to defend the evaluation, but it is likely that the reviewer did not understand parts of the paper

**Scope:**

4: The work is relevant to the Web and to the track, and is of broad interest to the community

---

### Official Review · Reviewer_fvaw · 2023-11-22

**Novelty:** 5
**Technical Quality:** 6

**Review:**

The paper focus on an interesting problem, how to learn robust representations and leverage information bottlenecks for dynamic graphs. The authors have shown that the proposed model Dynamic Graph Information Bottleneck framework can achieve good performance on learning robust and discriminative representations.
Pros:
1. The paper is well written and easy to follow.

2. The authors have well discussion of the bottleneck information and show promising results in experiments.

3. The authors conduct well designed experiments to show the power of DGIB, so it is very clear that the DGIB can achieve reasonable accuracy in both discriminative tasks and robust testing tasks.

Cons:
Some of the parameters should be clearly defined and discussed to avoid confusion.

**Questions:**

The setting of parameters should be discussed. For example, the setting of $\alpha$ and $\beta$, and how these two parameters influence the results. Furthermore, is there any connection between these parameters and the dataset properties?

How to define the 2 phases in training? Do the dataset properties influence the phase?

Thanks for the explanation, scores remain the same.

**Reviewer Confidence:**

4: The reviewer is certain that the evaluation is correct and very familiar with the relevant literature

**Scope:**

4: The work is relevant to the Web and to the track, and is of broad interest to the community

---

### Official Review · Reviewer_S7Jk · 2023-11-23

**Novelty:** 3
**Technical Quality:** 3

**Review:**

## Summary

This paper presents the  Dynamic Graph Information Bottleneck (DGIB) framework for dynamic graph representation learning. The framework is centrally based on the Minimal-Sufficiant-Consensual Condition, an assumption which relates the prediction of graph link with the mutual information between the signal propagated among the graph. The DGIB is empirically evaluated on various data and compared with various baselines.

## Pros

- The information theoretical approach to the dynamic graph representation learning is intriguing. A principle approach is always welcomed.
- Extensive baselines are selected for evaluating the performance of DGIB.
- The text is in general well written, and the figures are illustrative.

## Cons
- The core of DGIB, namely the MSC condition proposed in Assumption 1, is not well explained. It would be helpful if the authors can further explain the intuition behind this assumption, the motivation of such formulation, and the meaning of the MSC condition.
- The exact meaning of "consensual"in the context of DGNN is not explained. Is this related to the minimal sufficient statistics?
- Extensive derivations have been done based on the MSC condition. However, there lacks of supportive evidence, either theoretical or empirical, of why and to what extend this assumption should be true. This weakens the foundation of the whole work. The empirical evaluation of the whole DGIB provides merely indirect indications by showing the effectiveness of DGIB. However, since a major part of the contribution lies within the theoretical part, this is not enough for other researchers to follow this direction.
- The mathematical formulation somewhat hinders the readability and can be improved. Particularly, the abuse of superscript and subscript can be reduced. Examples includes the the $\mathcal{N}_{ST}(v, k, t)$ in equation (8) and the various minimization constrains written under the "min" operator. In general, more textual explanation is helpful to understand the motivation and thinking flow of each equation easier. Equations should serve for this purpose, instead of pursuing only accuracy and being written like programing codes.
- The link between the theoretical framework and the model robustness against adversarial learning is not clear (see Q1 below). And this hurts the relevance of DGIB's theoretical contribution.
- It would be both helpful and necessary to show the performance of DGIB on normal (non-adversarial) graph learning scenarios. DGIB will not be practically meaningful, if it cannot provide decent performance compared to the baselines in non-adversarial setting.
- Some related works should probably be discussed (see Q2 below).

**Questions:**

Q1: In the intro-section between roughly line 68 and 88, you explained the IB principle as an approach to improve model robustness against adversarial attack. However, as far as I know, the original papers from Naftali Tishby "analyze Deep Neural Networks (DNNs)  via the theoretical framework of the information bottleneck (IB) principle." and is a general theoretical framework to understand DNN from the information theoretical perspective. IB can indeed be used to explain generalization ability, model redundancy, and over-parameterization, with various works published. However, model redundancy or over-parameterization is, although somewhat related, not exactly the same as model robustness against adversarial attack, especially considering that adversarial attacks can be done from different perspectives and their information-theoretical formulation is unclear. Would you please explain further how IB, especially your DGIB, is related to model robustness theoretically? Providing references will also greatly help.

Q2: Structural Information (Structural Entropy) is a different yet related information-theoretical approach towards GNN. How is your work related to this series of works [1,2,3,4,5], or to Structural Information in general?

[1] Li, A., & Pan, Y. (2016). Structural information and dynamical complexity of networks. IEEE Transactions on Information Theory, 62(6), 3290-3339.

[2] Wu, J., Li, S., Li, J., Pan, Y., & Xu, K. (2022). A simple yet effective method for graph classification. arXiv preprint arXiv:2206.02404.

[3] Wu, J., Chen, X., Xu, K., & Li, S. (2022, June). Structural entropy guided graph hierarchical pooling. In International conference on machine learning (pp. 24017-24030). PMLR.

[4] Song, K., Yue, K., Duan, L., Yang, M., & Li, A. (2022, August). Mutual information based Bayesian graph neural network for few-shot learning. In Uncertainty in Artificial Intelligence (pp. 1866-1875). PMLR.

[5] Yang, Z., Zhang, G., Wu, J., Yang, J., Sheng, Q. Z., Peng, H., ... & Su, J. (2023, February). Minimum entropy principle guided graph neural networks. In Proceedings of the Sixteenth ACM International Conference on Web Search and Data Mining (pp. 114-122).

For other questions, please refer to the Cons.

**Reviewer Confidence:**

4: The reviewer is certain that the evaluation is correct and very familiar with the relevant literature

**Scope:**

3: The work is somewhat relevant to the Web and to the track, and is of narrow interest to a sub-community

---

### Official Review · Reviewer_4Guc · 2023-11-26

**Novelty:** 5
**Technical Quality:** 5

**Review:**

This paper focuses on dynamic graph representation learning, proposing the dynamic graph information bottleneck framework to learn robust and discriminative representations. Based on two assumptions, this paper claims that the optimal representations should satisfy the minimal-sufficient-consensual (MSC) condition. To meet the MSC condition, two decomposed objectives are further proposed to learn representation. Experiments are conducted on three real-world datasets.

Strength:
1. The studied problem, i.e., dynamic graph representation learning is important.
2. The perspective from the information bottleneck is interesting and makes sense.
3. Extensive experiments, including performance against random noise and adversarial attacks, information plane analysis, etc., are conducted.

Weakness:
1. The proposed methods are based on two assumptions, which lack further analysis, discussion, or support for the applicability or practicality of the proposed assumptions.
2. The necessity of the consensual condition is confused. While eq (4) and eq (6) are quite similar, why the consensus condition should be satisfied?
3. Lacking reference to or comparison with some related work [1], which also discusses and experiments the robustness of dynamic graph representation.
4. The writing of this paper needs to be further improved. For example, below eq (2), the authors explain H(*) which seems to be not shown in the paper.  The definition of N_{ST}(v,k,t) is confused for G^{t-1, t}.

[1] DyTed: Disentangled Representation Learning for Discrete-time Dynamic Graph. KDD 23

**Questions:**

1. Why the consensual condition is necessary? While eq (4) and eq (6) are quite similar, why the consensus condition should be satisfied?
2. Why from eq (7) to eq (9), C(\theta) has been directly replaced by \theta?
3. Why every Z^{t+1} is close to MS status for the next-step label Y^{t+1}, but fail to predict the final Y^{T+1}?
4. How about the applicability or practicality of the proposed two assumptions?

**Ethics Review Description:**

No ethical issues

**Reviewer Confidence:**

3: The reviewer is confident but not certain that the evaluation is correct

**Scope:**

4: The work is relevant to the Web and to the track, and is of broad interest to the community

---

### Decision · Program_Chairs · 2024-01-22

**Decision:**

Accept (Oral)

**Comment:**

This paper presents a novel Dynamic Graph Information Bottleneck (DGIB) framework to learn robust and discriminative representations to adversarial attacks. Based on two assumptions, this paper claims that the optimal representations should satisfy the minimal-sufficient-consensual (MSC) condition. To meet the MSC condition, two decomposed objectives are further proposed to learn representation. Experiments are conducted on three real-world datasets.

 The reviewers appreciate the work's novelty, method, experiments, and topic. The authors have addressed most of the reviewers' concerns during rebuttal.